# Assessing the spatial and temporal variability of GHG emissions from different configurations of on-site wastewater treatment system using discrete and continuous gas flux measurement

Jan Knappe [1], Celia Somlai [1], Laurence W. Gill [1]

[1] Department of Civil, Structural and Environmental Engineering, Trinity College Dublin, The University of Dublin, Ireland

*Correspondence to*: Laurence W. Gill (laurence.gill@tcd.ie)

**Abstract.** Global emissions linked to wastewater treatment are estimated to account for up to 1.5% of total greenhouse gas (GHG) emissions globally. However, few studies have measured GHG emissions from domestic on-site treatment systems (DWWTSs) directly. In this study, two DWWTSs were monitored for 446 days and >42,000 gas flux measurements were conducted using both discrete spot measurements and continuous flux chamber deployments. The observed GHG fluxes from biological activity in the soil and water phase were found to be highly spatially and temporally variable and correlated to environmental factors, water usage patterns and system design. In total, the results show that a septic tank discharging effluent into a well-designed soil treatment unit is estimated to emit a net 9.99 kg-$CO_2$eq cap$^{-1}$ yr$^{-1}$, with approximately 63%, 27% and 10% of the total $CO_2$-equivalent net emissions in the form of $CO_2$, $CH_4$ and $N_2O$, respectively. Emissions from the septic tank surface contributed over 50% of total emissions and tended to be strongly underestimated by one-off discrete measurements, especially when episodic ebullitive events are to be considered. Fluxes from the soil treatment unit (STU) stemmed from both the soil surface and the vent system. Soil fluxes were mostly influenced by temperature but peaked regularly under conditions of rapidly changing soil water content. Vent fluxes were mostly governed by effluent quality and a low number of high emission events was responsible for the majority of total observed vent emissions. Owing to the strong overall spatial and temporal heterogeneity of observed fluxes from DWWTSs across all modules, future studies should focus on continuous deployments of a number of flux chambers over discrete measurements to accurately assess GHG emissions from on-site systems. This study also provided insights into managing GHG emissions from DWWTSs by different system configuration design, as well as indicating that the current IPCC emission factors for $CH_4$ and $N_2O$ are significantly overestimating emissions for on-site wastewater treatment systems.

## 1 Introduction

Overall greenhouse gas (GHG) emissions from the waste and wastewater sector contribute an estimated 2% to the total national emissions in Ireland (EPA 2018). Global emissions linked to wastewater treatment are estimated to account for up to 1.5% of total GHG and 5% of non–$CO_2$ GHG emissions, and are expected to contribute 42% to all waste–related GHG emissions by 2030, compared to 36% in 1990 (Bogner et al. 2008; US EPA 2012). The quantification of direct GHG emissions from

wastewater treatment systems is currently based on the application of estimation methodologies that have been published by the Intergovernmental Panel on Climate Change (IPCC 2013). However, national and global estimations are considered highly uncertain as they are based on a limited number of case studies and rely heavily on secondary assumptions such as load-based calculations or emission factors rather than primary data.

Most research on GHG emissions from wastewater to date has been carried out on centralised, large-scale treatment systems (Cakir and Stenstrom 2005; Czepiel et al. 1993; Johansson et al. 2004; Yver Kwok et al. 2015; Masuda et al. 2015; Yoshida et al. 2014). Few studies, however, have directly measured GHG emissions from decentralised and/or on-site wastewater treatment systems (Diaz-Valbuena et al. 2011; Huynh et al., 2021; Leverenz et al. 2011; Somlai-Haase et al. 2017; Somlai et al. 2019; Truhlar et al. 2016; Wigginton et al. 2020), despite an estimated 20% of the population relying on on-site wastewater treatment in the European Economic Area and the United States (EEA 2013; US EPA 2016). The proportion of these systems is significantly higher in low– and middle– income countries in comparison to the global average, reckoned to provide improved sanitation for up to 64% of such countries populations (Blackett et al. 2014).

Domestic on-site wastewater treatment systems (DWWTSs) are environmentally and economically sustainable, small scale, decentralised systems usually comprising a septic tank (ST) for collection, storage and partial treatment of effluent, sometimes followed by a manufactured secondary treatment unit, which then discharges into some form of soil treatment unit (STU) (Cooper et al. 2016; Gill et al. 2009). DWWTSs are suitable for off–the–grid solutions as they fundamentally rely on naturally occurring biogeochemical processes for the treatment of wastewater. Despite their apparently straightforward design, the biogeochemical processes involved are complex and involve a wide range of microbial populations existing in different redox conditions (Beal et al. 2005; Tomaras et al. 2009). Two main redox environments are usually defined for on–site systems (Wilhelm et al., 1994): the first zone in the ST is an anaerobic environment where high concentrations of organic C is mainly degraded via hydrolysis, acidogenesis, and methanogenesis producing $CO_2$ and $CH_4$; the second redox zone is the STU in which both aerobic and anaerobic conditions usually exist. The oxygen required for aerobic oxidation of organic C in the partially treated effluent (and consequent production of $CO_2$) is supplied by gaseous diffusion within the unsaturated zone.

In STU trenches, a biomat forms at the infiltrative surface with time and gradually clogs soil pores. The development of this biomat layer is linked to the physical accumulation of suspended solids within soil pores and microbial growth (McKinley and Siegrist 2010; Thomas et al. 1966; Beach et al. 2005; Siegrist and Boyle 1987; Knappe et al. 2020; Jones and Taylor 1965; Bouma 1975). As gradual biomat development results in a growing resistance to flow and reductions in hydraulic conductivity, ponding of effluent at the trench base can occur and anaerobic conditions may develop (Beach et al. 2005; Hu et al. 2007; Knappe et al. 2020; Siegrist and Boyle 1987; Van Cuyk et al. 2001). Within these ponded infiltration trenches and mature biomats, as well as other areas of the STU which lack oxygen such as saturated microsites, anaerobic organisms (methanogenic bacteria etc.) act to break down insoluble organic compounds into $CO_2$ and $CH_4$. Furthermore, it has been suggested that elevated concentrations of $CH_4$ in the soil above the infiltration trenches can host a population of methanotrophs, which may function to reduce overall $CH_4$ fluxes from the STU to the atmosphere (Fernández-Baca et al., 2018).

In this study two operational DWWTSs were instrumented and monitored for more than a year in order to provide the first quantification of total net GHG emissions (including $CO_2$, $CH_4$ and $N_2O$ measurements from the septic tank, the STU and the vent system). The STUs at both DWWTSs received effluent of two different strengths (primary treated effluent high in organics and $NH_4$-N as well as secondary treated effluent with lower organic strength but high in $NO_3$-N), enabling the impact of pre-treatment on GHG emissions from the soil to be directly compared.

## 2 Study Design and Methods

### 2.1 Research Sites

Two new DWWTSs were constructed as research sites serving single detached houses in Co. Limerick, Ireland. Site A, a pumped-flow system, was constructed in 2015 and Site B, a gravity–flow based system, was constructed in 2016. Both sites were instrumented with inserts for gas flux measurements within the four-trench STU and undisturbed control soil, a weather station, and a network of soil sensors for monitoring volumetric water content (VWC). Both systems consisted of a two-chamber prefabricated concrete septic tank (Aswasep Septic Tank NS4S, Molloy Precast, Ireland), each with a capacity of 4760 L. Following the ST, one half (two trenches) of the STU was directly fed with primary effluent (PE) from the ST, while the other half (two further trenches) of the STU was fed with secondary treated effluent (SE) that underwent additional treatment by a media filter (coconut husk) or rotating biological contactor (RBC) in Site A and B, respectively. The RBC is a fixed film, secondary wastewater treatment process in which plastic discs slowly rotate bring the attached biofilm down into the sewage (the substrate) and then up into the air (for oxygen transfer). Splitting the effluent between primary and secondary fed STU trenches allowed for the direct comparison of GHG emissions from STUs receiving effluent of different pre–treatment levels under identical subsoil, meteorological, and environmental conditions. The research sites, soil characteristics, and water quality data over a two year intensive monitoring period were reported in detail in Knappe et al. (2020) A short characterisation of the research sites is given in Table 1.

**Table 1:** Overview of site characteristics and installed system as reported in Knappe et al. (2020).

| Parameter | Site A | Site B |
| --- | --- | --- |
| Location | Kilmallock, Co. Limerick | Crecora, Co. Limerick |
| Subsoil type | Sandy Loam | Loam |
| Sand – Silt – Clay | 59% – 30% – 11% | 49% – 34% – 17% |
| Bulk Density | 1.44 g cm$^3$ | 1.20 g cm$^3$ |
| Porosity | 0.386 | 0.448 |

| | | |
|---|---|---|
| Saturated hydraulic conductivity | 30.9 cm d$^{-1}$ | 13.9 cm d$^{-1}$ |
| Construction | September 2015 | April 2016 |
| Primary treatment | Septic tank | Septic tank |
| Secondary treatment | Cocopeat media filter | Rotating biological contactor |
| Flow regime | Pumped flow | Gravity flow |
| Number of occupants | 5 | 4 |

## 2.2 GHG flux measurements

### 2.2.1 Equipment and flux calculation

An integrated and automated soil gas flux measurement systems for $CO_2$ (LI-8100A, Li-Cor Biosciences, Inc.) and $CH_4$ (UGGA 915-0011, Los Gatos Research) was employed to measure concentrations from gas flux chamber inserts, the vent system, and dissolved gases in the effluent. A detailed description of the measuring system used in this research was published in Somlai-Haase et al. (2017). Due to temporary instrument failure of the $CH_4$ analyser, more measurements are available in general for $CO_2$ compared to $CH_4$ fluxes. Additional gas samples were collected manually from the sampling loop through a Tee-fitting with a septum (8100-664 Trace Gas Sample Kit, Li-Cor Biosciences, Inc.) for analysis of $CH_4$ and $N_2O$ by gas chromatography in the laboratory at Trinity College Dublin. For that, 50 ml of gas was extracted from the gas loop and transferred into sealed, pre-evacuated 20 ml vials for transport. Samples were, then, injected and analysed within 72 hours using a gas chromatograph (Clarus 500, Perkin Elmer) equipped with capillary columns (Elite-Plot Q), a flame ionisation detector for $CH_4$, and electron capture detector for $N_2O$.

Fluxes were calculated using a mass balance approach as described in Somlai-Haase et al. (2017) and also in the Supplemental Information. Observations resulting in model fits with low overall $R^2$–values ($R^2 < 0.9$ for $CO_2$ and $R^2 < 0.8$ for $CH_4$ and $N_2O$) were assumed compromised and discarded, representing a total of 1%, 21% and 3% of $CO_2$, $CH_4$ and $N_2O$ measurements, respectively. Flux values were, then, scaled and converted into a mass flux of gas per capita (expressed in g cap$^{-1}$ d$^{-1}$) using the ideal gas law and household occupancy levels.

### 2.2.2 Fluxes from the septic tank

Gas fluxes from the ST were evaluated using discrete and continuous measurements. Discrete spot survey measurements were carried out during each site visit (n = 14 and 13 for $CO_2$ for Site A and B, respectively; n = 9 and 8 for $CH_4$ for Site A and B, respectively) by placing the survey chamber (LI-8100 103, Li-Cor Biosciences, Inc.) on fixed inserts standing on submerged

tripods in each of the two ST chambers for a period of five minutes at a time (see Supplementary Information for details). Further continuous measurements (> 18 hours) were carried out over the secondary ST chamber once per site using the automated survey chamber mounted on the same fixed insert in order to assess the diurnal variability of observed fluxes (Site A: 24/25 July 2018; Site B: 20/21 November 2017). To quantify dissolved concentrations in the effluent, grab samples (100 ml) were collected from both ST chambers and transferred into a bubbler bottle and connected to the gas analyser (see Supplementary Information for details). Effluent temperature, electrical conductivity (EC), pH, and dissolved $O_2$ were recorded simultaneously using a multiparameter kit (ProfiLine Multi 3320, WTW GmbH, Germany) and dissolved $CO_2$ and $CH_4$ concentrations were estimated using the headspace method (Hope et al. 1995).

### 2.2.3 Fluxes from the STU

A total of 20 inserts were permanently installed into the soil at random locations over the first 6 m of each STU area (Figure 1). Some of these were located directly above the gravel filled trenches (above which the original topsoil had been replaced after being excavated), with the other inserts located above undisturbed soil within the STU areas A further four inserts were located over undisturbed soil away from the area of the STUs to act as the best available controls in both sites. All inserts were installed to a depth of approximately five cm with another five cm clearance above ground on which flux chambers could be placed. Both discrete and continuous measurements were used to quantify gas fluxes from the STU. Discrete spot measurements were carried out sequentially during each site visit (n = 13 and 15 for $CO_2$ at Site A and B, respectively; n = 5 and 8 for $CH_4$ at Site A and B, respectively) to quantify the spatial distribution of fluxes over the STU by placing the survey chamber for an incubation period of three minutes over each of the 24 inserts. In between site visits, continuous flux measurements were carried out over four randomly chosen soil inserts (one per trench) within the STU and one control. Measurements over each insert were automatically taken once an hour with a six minutes incubation period. The deployment of continuous measurements was alternated between both sites and left in place until the next site visit, resulting in continuous diurnal flux time series of between 14 to 37 consecutive days. Discrete $N_2O$ fluxes were only measured on 8 occasions on each site, due to instrument availability and increased manual handling effort.

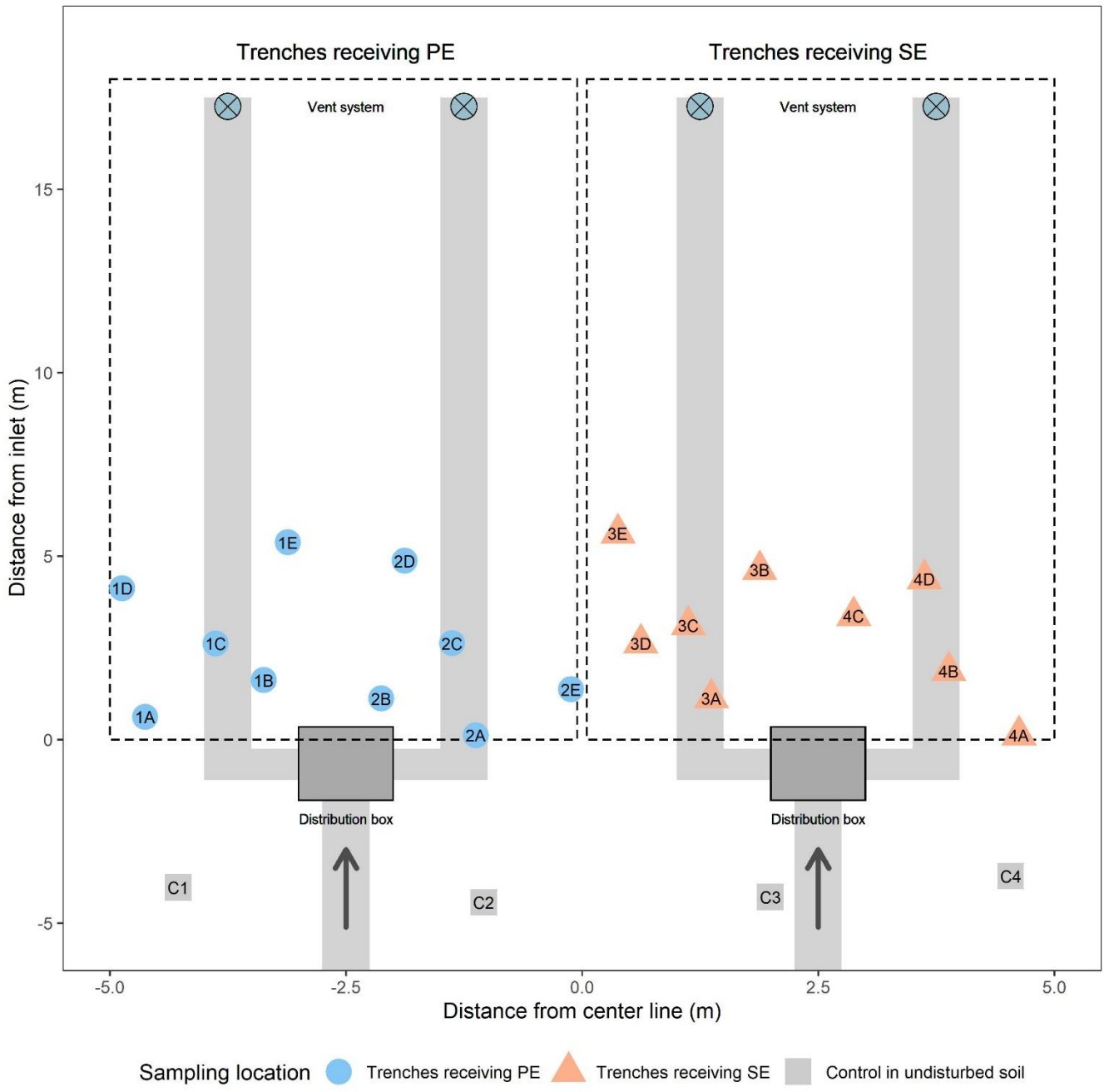

**Figure 1:** Positions of permanently installed inserts for gas flux measurements within the STU over trenches receiving primary effluent (PE; blue circles) and secondary effluent (SE; red pyramids) and in undisturbed soil (control; yellow squares) on both sites. The effluent inflow is marked by arrows. Subsurface trenches are marked in gray. Vents are marked by crosses on blue circles.

### 2.2.4 Fluxes from the vent system

Perforated effluent distribution pipes embedded in the STU gravel trenches were connected to above–surface vents by elbows
fitted at the end of each pipe. Gases escaping through these vents were captured using a Vent Wizard 800+ (in–house development; see Supplementary Information) consisting of a sealed end cap with a single gas line connected to the gas sampling loop. Before capping, the average temperature and undisturbed air velocity inside the vent were determined using a hotwire anemometer (LU8050, TQC Sheen, NL). Subsequently, gas concentrations inside the capped vent system were monitored for at least 3 min and until no further increase was observed. Gas fluxes from the vent were then determined by
scaling the final steady–state gas concentration observed inside the vent with the surface area of the vent port and determined air flow velocity inside the vent (n = 9 and 8 for $CO_2$ for Site A and B, respectively; n = 4 and 3 for $CH_4$ for Site A and B, respectively). Additional samples for $N_2O$ were collected at 6 occasions from each site.

### 2.3 Auxiliary data

Meteorological data were recorded at both sites using a weather station (Campbell Scientific Ltd, UK). Mean air temperature,
barometric pressure, net radiation, rainfall, relative humidity, wind direction, and wind speed was used to calculate Penman-Throughout the study the pollutant and hydraulic loads in the systems were monitored and presented in detail in Knappe et al. (2020). The mean daily wastewater production for each household was 287 L $d^{-1}$ and 500 L $d^{-1}$ at Site A and B respectively, of which approximately one fourth was delivered into each of the four STU trenches. While the overall pollutant loading out of the ST in Site A was generally lower as compared to Site B, the RBC installed on Site B performed considerably better
when it came to removing organics and TN than the media filter in Site A.

Effluent distribution inside the STU trenches was traced using network of soil sensors (type EC5 and GS3, Decagon Devices, Inc., USA) installed within the STU which provided detailed information on VWC, soil temperature and pore water EC. The relatively higher organic loading in trenches receiving PE on both sites caused the development of an extensive biomat, spreading approximately 15 m along the trenches after three years of operation. The biomat growth in the trenches receiving
SE did not extend to more than 7.5 m and 10 m at Site A and Site B respectively, reflecting more limited biomat growth due to the lower availability of organics and nutrients in the effluent feeding these trenches (Knappe et al., 2020). The percolation of the effluent through the soil caused a significant reduction in organic C concentrations within the first 300 mm depth below the infiltrative surface (Dubber et al., 2021) and net N removal more effective in trenches receiving PE as also found in previous studies of STUs (e.g. Beal et al., 2005; Gill et al., 2009).

### 2.4 Data collection and statistical analysis

All on-site sensor data was collected hourly on a CR1000 data logger (Campbell Scientific Ltd, UK). Data analysis was carried out using R, version 3.6.2 (R Core Team 2019). Statistical significance of between-treatment medians was calculated using Wilcoxon signed-rank tests and is reported using $p$ values or adjusted $p_{adj}$ values with Bonferroni correction (in case of

multiple comparisons). Substantive significance is reported as standardised effect sizes $\bar{r}$, a relative measure for the magnitude of the difference between groups (Sullivan and Feinn 2012). The significance level $\alpha = 0.05$ was used to test all hypotheses. Bias-corrected and accelerated bootstrap 95% confidence intervals were determined around the medians and effect sizes using 5000 resampling draws (Carpenter and Bithell 2000; Banjanovic and Osborne 2016). Confidence intervals are reported in brackets next to the point estimate. Where applicable, correlations are evaluated using Pearson's $r$ coefficient.

## 3 Results and Discussion

### 3.1 Total number of observations

To assess the spatial and temporal variability of GHG emission from two DWWTSs, 42,198 gas flux measurements over the course of 446 days in 2017 -2018 were collected using discrete spot measurements and continuous deployments (Figure S1). A total of 781, 382 and 62 observations were measured as discrete spot measurements throughout the study period for $CO_2$, $CH_4$ and $N_2O$, respectively (Table 1), as well as 34,660 and 6,313 observations during continuous deployments of automated gas flux chambers for $CO_2$ and $CH_4$, respectively. The total number of $CH_4$ measurements was lower than those for $CO_2$ mainly due to limited instrument availability due to instrument breakdown. Approximately 2.5% of the observations were recorded at the ST surface, 0.5% at the STU vent system and the remainder over the STU from the soil gas flux measurements, mainly during the automated long-term deployments.

### 3.2 Fluxes from the ST

As the effluent passes through the anaerobic environment of the two-chamber ST, microbiological degradation of organic matter via hydrolysis, acidogenesis, and methanogenesis produces $CO_2$ and $CH_4$. No significant production of $N_2O$ would be expected in the ST as there is almost no nitrate present in incoming raw sewage and the prevailing environment would not support the prerequisite oxidation of organic-N and ammonium. Fluxes were directly measured from the ST surface in both chambers using fixed collars and discrete measurements.

Discrete flux measurements inside the ST revealed that fluxes of $CO_2$ from the first chamber were generally higher than fluxes from the second chamber (Figure S2A). Median $CO_2$ emission from the first chamber (Md = 3.50 µmol $CO_2$ m$^{-2}$ s$^{-1}$) were generally higher as compared to the second chamber (Md = 2.63 µmol $CO_2$ m$^{-2}$ s$^{-1}$) and expressed peak values of up to 18.52 and 7.85 µmol $CO_2$ m$^{-2}$ s$^{-1}$, respectively (Table S2), which could be attributed to the fact that flow conditions within the first chamber are generally expected to be more turbulent due to pulses of incoming effluent during periods of peak flow. This difference was significant with moderate effect size (p = .016; $\bar{r}$= .48 [.13; .76]) and was corroborated by, i) slightly higher dissolved $CO_2$ concentrations found in effluent samples from the first (Md = 18.65 mg $CO_2$ L$^{-1}$) as compared to the second

chamber (Md = 12.98 mg $CO_2$ $L^{-1}$) and ii) a significant positive correlation between observed $CO_2$ fluxes across both chambers with respective wastewater TOC concentration (p < .001; $r$ = .51). This indicates that the release of $CO_2$ from the ST surface was mainly driven by the availability of organic matter as opposed to temperature which was not correlated with $CO_2$ fluxes (p = .29; $\bar{r}$ = .15).

On the contrary, measurements of discrete $CH_4$ fluxes inside the ST revealed that fluxes were slightly higher in the second chamber as compared to the first chamber (Figure S2B). Median $CH_4$ emissions were 0.21 and 0.34 µmol $CH_4$ $m^{-2}$ $s^{-1}$ in the first and second chamber, respectively. However, the first chamber expressed higher variability with peaks of up to 1.86 and 1.07 µmol $CH_4$ $m^{-2}$ $s^{-1}$ (Table S1), which again could be attributed to the fact the flow conditions within the first chamber are generally expected to be more erratic. The overall difference between both chambers was not significant and the estimated effect size was small, albeit with a wide confidence interval (p = .76; $\bar{r}$= .09 [.01; .65]). Median dissolved $CH_4$ concentrations followed this pattern and were 0.25 and 0.41 mg $CH_4$ $L^{-1}$ in the first and second chamber, respectively. $CH_4$ fluxes from the ST neither correlated with organic content (p = .92; $\bar{r}$ = -.02) nor temperature (p = .16; $\bar{r}$ = .26) indicating that methane production becomes more prevalent as the microbiological degradation of organic matter progresses in the ST. Initial measurements of $N_2O$ did not yield detectable fluxes from either chamber and were thus discontinued after three occasions.

Diurnal measurements

To assess the representativeness of the one-off discrete measurements, continuous measurements of fluxes from the second chamber were performed on each site. On Site A, 280 flux measurements were taken within 24 hours in July 2018. On Site B, 220 flux measurements were taken within 19.5 hours in November 2017.

While Site A expressed high fluctuations of observed fluxes with a considerable number of ebullitive events for both $CO_2$ (CV 0.49) and $CH_4$ (CV 0.68), fluxes in Site B expressed more gradual changes in observed fluxes over the course of the day without distinct ebullitive events for both $CO_2$ (CV 0.13) and $CH_4$ (CV 0.19) (Figure 2A, Table S3). Both sites expressed a distinct diurnal behaviour with 43.1% and 0.5% of observed values measured during the continuous chamber deployment falling outside the range of fluxes observed during the discrete chamber deployment on Site A and B, respectively (Figure 2A). The lack of distinct ebullitive events in Site B could be linked to the overall lower mean ambient temperatures during the trial in November for Site B (12.5 °C) as compared to July for Site A (18.1 °C), as colder months are marked by overall lower microbial activity and higher gas solubility, thus lowering the potential for detecting gas fluxes from the surface. Additionally, the presence of a thick scum layer (approx. 30 cm) that had accumulated within the first chamber of the ST in Site B effectively reduced overall turbulence caused by peak flow events. On Site A, however, periods of both high $CO_2$ and $CH_4$ fluxes were clearly correlated with water usage patterns and peaks occurring in the morning, around noon and again in the evening hours (Figure 2A), indicating that the hydraulic disturbance in the ST must have led to degassing of dissolved gases and dislodgement of entrapped gas bubbles that had built up in the tank.

Comparing flux values obtained by continuous and discrete measurements shows that observed fluxes obtained by discrete measurements were significantly lower (p < .001 for $CO_2$ and p = .032 for $CH_4$), indicating that discrete measurements underestimated fluxes from the ST surface by a factor of up to two for median fluxes and up to seven for peak fluxes (Figure 2B). These results suggest that flux estimates derived from discrete measurements alone were generally not sufficient to capture the full range of fluxes occurring from a ST. Discrete measurements tended to underestimate median GHG fluxes by up to

2.75 μmol $CO_2$ m$^{-2}$ s$^{-1}$ and 0.77 μmol $CH_4$ m$^{-2}$ s$^{-1}$ on each site, especially when episodic ebullitive events are to be considered. This matches findings by researchers investigating methane emissions from freshwater streams, lakes and reservoirs (Bastviken et al., 2011; Natchimuthu, 2016).

The measurement of dissolved $CO_2$ and $CH_4$ concentrations in both chambers of the STs broadly reflected the gas fluxes: higher $CO_2$ concentrations in chambers 1 than 2, compared to higher $CH_4$ concentrations in chambers 2 than 1 (Table S4). The

difference between $CO_2$ and $CH_4$ fluxes and dissolved concentrations across both chambers might be explained by the compartmentalisation of subsequent stages of low rate anaerobic digestion in the ST: in the first stage, organic matter is converted to simple organic compounds and volatile fatty acids producing mostly $CO_2$ and in the second stage, the soluble organic acids (which will have leached into the bulk liquid and passed into the second chamber) are stabilised by methanogens and most of the $CH_4$ is produced. A parallel study into sludge accumulation in the STs on these sites (Gill et al., 2018), showed

that the first chambers of the STs had approximately 5 times higher sludge mass accumulation compared to the second chambers after 2 years of operation.

Studies measuring GHG emissions from ten STs receiving only black water in Vietnam using a similar floating chamber method found significant emissions of both $CO_2$ and $CH_4$ (7.39 and 91.37 kg cap$^{-1}$ yr$^{-1}$, respectively) in the first chamber while $N_2O$ emissions were negligible (Huyn et al., 2021). Notably, STs that have not been desludged within the preceding five years

of the study expressed significantly higher $CH_4$ emissions as compared to regularly desludged tanks. While the systems in our study were <5 years old at the time of sampling, it is not uncommon to observe desludging intervals of 5 years or higher in Ireland (Mac Mahon et al., 2022). For such tanks it is likely that >40% of the tank volume is occupied by sludge which means a denser sludge layer at the bottom with longer solids retention time allowing enough time for the relatively slow growth of methanotrophs and conditions to promote anaerobic digestion.

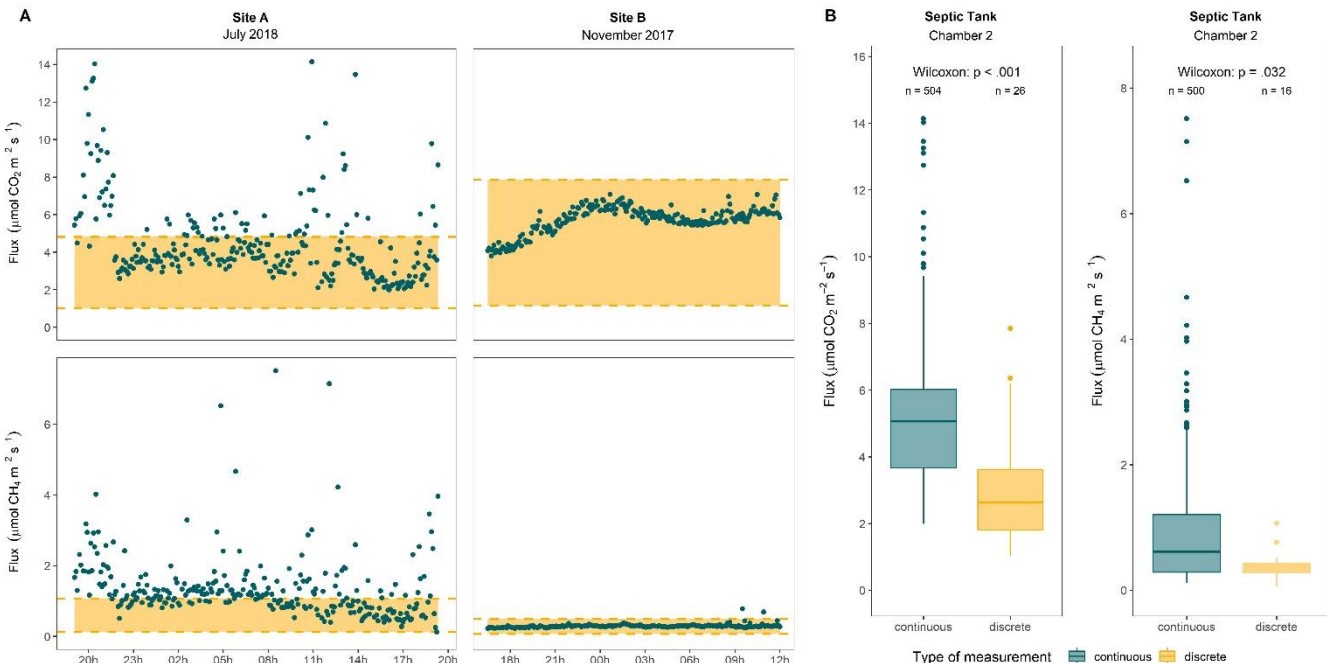

**Figure 2:** $CO_2$ and $CH_4$ fluxes from the ST. (a) Comparison of the time series of observed fluxes from the second chamber of STs in Site A and B during overnight deployment (blue circles) with range of fluxes observed during discrete measurements (red shaded area). Dashed lines represent respective minimum and maximum flux values as observed during discrete measurements which were usually taken in the late morning or early afternoon. (b) Boxplots of observed fluxes for $CO_2$ and $CH_4$ over the ST second chamber measured with continuous and discrete chamber deployment. Statistical results are presented as p-value of Wilcoxon signed rank tests with estimated effect size r and corresponding boot-strapped 95% confidence intervals; n denotes the number of observations per group.

### 3.3 Fluxes from the STU

#### 3.3.1 Discrete measurements

To assess the spatial variability of GHG fluxes from the STU a total of 1017 discrete gas flux measurements were performed over 20 permanently installed inserts in the STU (plus four control inserts over undisturbed soil) of which approximately 65% were conducted for assessing $CO_2$, 30% for $CH_4$ and the remainder for $N_2O$ fluxes. Measurements were taken on 13 and 15 sampling dates for Site A and B, respectively, over a period of 15 consecutive months (Figure S1). Discrete measurements were usually taken during late morning or early afternoon.

Discrete measurements revealed that fluxes of $CO_2$ above the trenches in the STU receiving PE (Md = 3.00 [2.70; 3.38] μmol $CO_2$ m$^{-2}$ s$^{-1}$) were slightly lower ($p_{adj}$ = .33; $\underline{r}$ = .07 [.00; .19]) than those fluxes measured at the Control (Md = 3.06 [2.51; 3.62] μmol $CO_2$ m$^{-2}$ s$^{-1}$) but not significantly. Discrete measurements of $CO_2$ above the trenches receiving SE (Md = 2.73 [2.40; 3.04] μmol $CO_2$ m$^{-2}$ s$^{-1}$) were significantly lower than those above the Control ($p_{adj}$ = .02; $\underline{r}$ = .18 [.05; .30]). The $CO_2$ fluxes from above the PE trenches were significantly higher than the fluxes above SE trenches ($p_{adj}$ = .048; $\underline{r}$ = .15 [.02; .28]). With respect to $CH_4$, there were almost no detectable fluxes measured above the STU trenches receiving either PE (Md = 0.002 [-0.001; 0.004] nmol $CH_4$ m$^{-2}$ s$^{-1}$) or SE (Md = -0.001 [-0.003; 0.001] nmol $CH_4$ m$^{-2}$ s$^{-1}$) with no significant difference between them ($p_{adj}$ = .44; $\bar{r}$= .08 [.00; .26]).  Although the $CH_4$ fluxes above both the trenches receiving PE and SE were significantly higher than the fluxes measured at the Control ($p_{adj}$ = .02; $\bar{r}$= .28 [.08; .46] and $p_{adj}$ = .002; $\bar{r}$= .36 [.16; .52]) respectively, it should be noted that that $CH_4$ fluxes from the control areas at both sites were negative or non-detectable throughout all discrete measurements (Md = -0.004 [-0.008; 0.004] nmol $CH_4$ m$^{-2}$ s$^{-1}$) and ranged from -3.36 to 0.01 nmol $CH_4$ m$^{-2}$ s$^{-1}$. Hence, the undisturbed control soils acted as a $CH_4$ sink which is in line with previous findings from soils of grasslands and unfertilised pastures, e.g. reported by Mosier et al. (1991, 1997), Dunfield et al. (1995), and Braun et al. (2013).

Discrete measurements of $N_2O$ fluxes above the trenches receiving PE (Md = -0.02 [-0.06; 0.03] nmol $N_2O$ m$^{-2}$ s$^{-1}$) and SE (Md = 0.05 [0.01; 0.09] nmol $N_2O$ m$^{-2}$ s$^{-1}$) were slightly lower than those fluxes measured at the Control  (Md = 0.06 [-0.02; 0.08] nmol $N_2O$ m$^{-2}$ s$^{-1}$) but not significantly so ($p_{adj}$ = .33; $\underline{r}$ = .32 [.01; .61] and $p_{adj}$ = .33; $\underline{r}$ = .26 [.01; .57] respectively). There was no significant difference between the $N_2O$ fluxes measured above the PE trenches compared to the SE trenches ($p_{adj}$ = .95; $\underline{r}$ = .01 [.00; .03]).

### 3.3.2 Spatial distribution

The spatial distribution of the net fluxes was analysed, first as a function of distance from the inlet pipes into the STU, as shown on Figure 3A. A comparison of the regression lines, whilst each is not significantly different from zero, doesshow that $CO_2$ fluxes above the section of the STU receiving PE seem to increase with distance, compared to only a very slight increase in emissions above the SE trenches with distance. In comparison, $CH_4$ emissions above the PE trenches, whilst again not found to be statistically significant, decrease with distance compared to almost no change with distance above the SE side of the STU. Equally, no change with distance is revealed for the $N_2O$ fluxes above either PE or SE side of the STU

The spatial distribution of the net fluxes was then analysed with respect to the lateral position of where the emissions were measured, whether located directly over the percolation trenches or between the trenches in the STU, as shown on Figure 3B. For the STU receiving PE, higher net $CO_2$ fluxes were measured from positions located between trenches (Md = 0.53 [0.34; 0.84] μmol $CO_2$ m$^{-2}$ s$^{-1}$) as compared to inserts located above trenches (Md = -0.32 [-0.48; -0.07] μmol $CO_2$ m$^{-2}$ s$^{-1}$). The same pattern was found for the SE side of the STUs with higher net fluxes between trenches (Md = 0.27 [0.00; 0.42] μmol $CO_2$ m$^{-2}$ s$^{-1}$) as compared to inserts located above trenches (Md = -0.71 [-0.92; -0.51] μmol $CO_2$ m$^{-2}$ s$^{-1}$). Both differences are

significant (p < .001) and have a moderate effect size ($\bar{r}$ = .26 [.26; 38] and $\bar{r}$ = .32 [.33; 43] for PE and SE trenches, respectively). This indicates that some $CO_2$ being generated in and below the trenches must be making its way to the atmosphere back up through the gravel and the percolation pipes via the vents (and not up through the soil), as quantified and discussed later. For $CH_4$ the median net fluxes over inserts located between or over gravel trenches for both trenches receiving PE and SE were both zero. However, the presence of a sufficient number of high emission events combined with a near complete lack of uptake events captured during discrete measurements rendered the flux observed directly over trenches (Md = 0.01 [0.00; 0.60] nmol $CH_4$ $m^{-2}$ $s^{-1}$) significantly different (p = 0.008) from fluxes observed between trenches. For $N_2O$ there were generally very low net fluxes for between and directly over trenches. Due to the limited sample size, no statistical difference was detected. The very low net fluxes suggest that nitrogenous compounds in the effluent in the form of ammonia and organic nitrogen are being transformed via nitrification into soluble nitrate (which percolates down to the underlying groundwater) and/or complete dentification or anammox to generate $N_2$ gas as the final product (Gill et al., 2009).

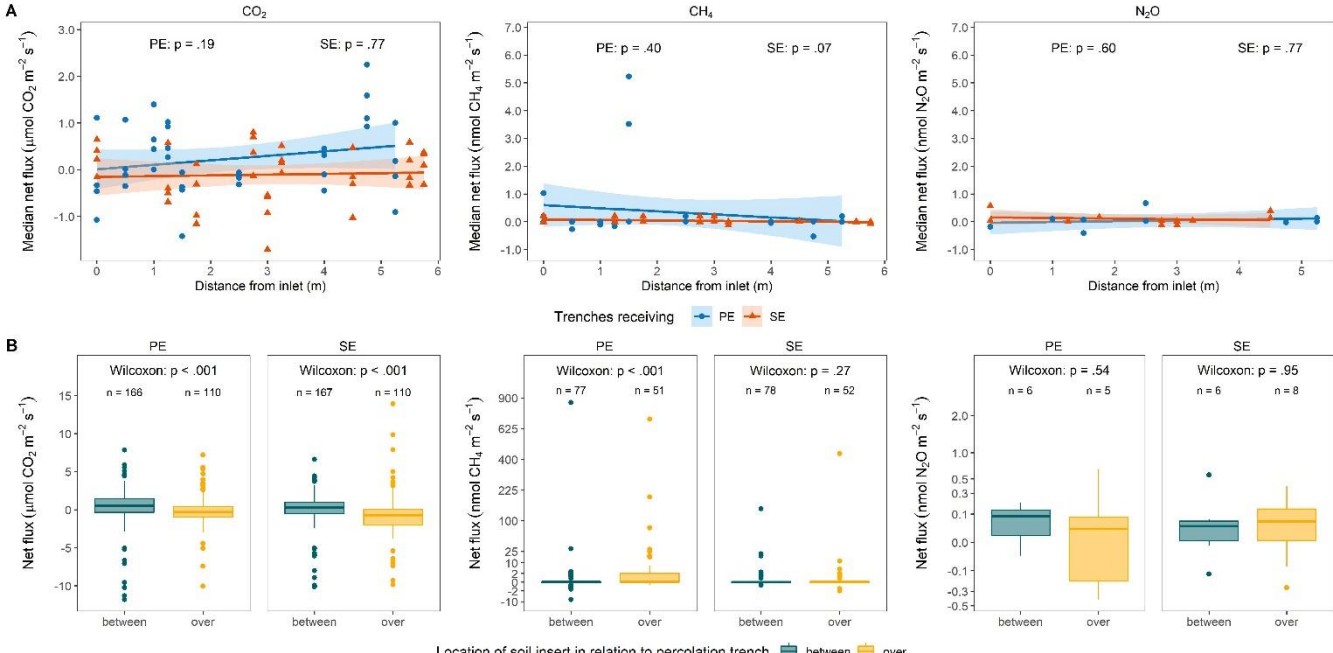

**Figure 3:** (A) Quarterly median fluxes observed by discrete GHG measurements over the STU as a function of distance from the inlet for trenches receiving PE (blue circles) and SE (red pyramids). Linear fits are marked by solid lines. The shaded areas represent the standard error of the fit. (B) Boxplots comparing net fluxes observed by discrete GHG measurements over the STU from soil inserts located either between (blue) or over (red) gravel trenches. Statistical results are presented as p-value of Wilcoxon signed rank tests; n denotes the number of observations per group. Note that in both subfigures the y-scale for $CO_2$ is given in units different from units used for $CH_4$ and $N_2O$ and that the y-axis for $CH_4$ and $N_2O$ has been square-root transformed in subfigure B to improve data visualisation.

### 3.3.3 Continuous measurements

To assess the temporal variability of GHG fluxes from the STU automated gas flux measurements were performed at hourly intervals over a set of four of the 20 permanently installed inserts in the STU (plus one of the four control inserts over undisturbed soil). Measurements were taken over the course of a total of 154 and 195 days for Site A and B, respectively.

During the continuous deployment of gas flux chambers, the undisturbed control soil emitted a median flux of 5.10 µmol $CO_2$ $m^{-2}$ $s^{-1}$ which was significantly more ($p_{adj}$ = .001) than the STU for both trenches receiving PE (Md = 2.17 µmol $CO_2$ $m^{-2}$

$s^{-1}$) and SE (Md = 2.05 µmol $CO_2$ $m^{-2}$ $s^{-1}$) – see Table S5. This finding which matches the pattern from the discrete measurements, suggests that either the architecture of the STUs must be providing an alternative pathway for the gases being generated by the microbial breakdown of the organics in the effluent (as well as gases generated by more natural soil processes) to get to the atmosphere, and/or, less comprehensibly, that the addition of organic effluent into the STU area is somehow promoting conditions for the soil to act as a net $CO_2$ sink possibly by changing the microbial diversity within the soil leading

to accumulation of organics. . It should also be noted that sections of the STU where the pipes are located have had 500 mm of soil replaced by gravel, missing soil compared to the control sites that would be contributing to $CO_2$ fluxes if present. The relative effect size of both differences was moderate with 0.59 and 0.50 for PE and SE, respectively. The difference between pre-treatment levels was small, but significant ($p_{adj}$ < .001).

For $CH_4$, the undisturbed control soil expressed a net median uptake of -0.42 nmol $CH_4$ $m^{-2}$ $s^{-1}$, a significantly higher uptake

than observed over the STU for both trenches receiving PE (Md = -0.30 µmol $CH_4$ $m^{-2}$ $s^{-1}$, $p_{adj}$ < .001) and trenches receiving SE (Md = -0.37 µmol $CH_4$ $m^{-2}$ $s^{-1}$, $p_{adj}$ = .003) even though both STUs were also net sinks,- see Table S5. The relative effect size of both differences was moderate to low with 0.40 and 0.10 for PE and SE, respectively, and the difference between pre-treatment levels was relatively small in absolute terms, yet significant ($p_{adj}$ < .001), with a moderate effect size of 0.41. Despite the overall range of fluxes observed from the control soil being comparable to the range of fluxes from PE trenches,

only 0.3% of control fluxes were higher than 2 µmol $CH_4$ $m^{-2}$ $s^{-1}$, compared to 14.6% and 2.0% for PE and SE trenches, respectively. This suggests that the relatively high abundance of high emissions events over PE trenches must have been the main driver of the difference between PE and SE trenches, emphasising the importance of capturing the entire range of potential fluxes.

### 3.3.4 Diurnal patterns

$CO_2$ fluxes expressed distinct diurnal variations with median peaks in the early afternoon at 15h, 14h and 12h for the control soil and PE and SE trenches, respectively (Figure 4A). The lowest median $CO_2$ fluxes occurred during the early morning hours at 5h, 6h and 6h for the control soil and PE and SE trenches, respectively. This pattern implies a strong dependence of $CO_2$ fluxes on diurnal temperature variations, independent of treatment, with peaks occurring at approximately the same time (5h and 15h for minimum and maximum mean temperature, respectively) which is, in turn, correlated with microbial activity. It

has to be noted, however, that control fluxes were most strongly affected by mid-day positive flux peaks as compared to STU fluxes, which are probably damped by some the gas finding an alternative route to the atmosphere along the trenches and up through the vent pipes, as mentioned above. $CH_4$ flux, on the other hand, expressed only weak to no diurnal patterns (Figure 4A), indicating that other factors than temperature, such as substrate availability, methanotrophic activity or changes in water content, must have been the main drivers responsible for $CH_4$ fluxes as suggested by, for example, Fernández-Baca et al.

(2018), Somlai et al. (2019), Swenson et al. (2019) and Truhlar et al. (2019).

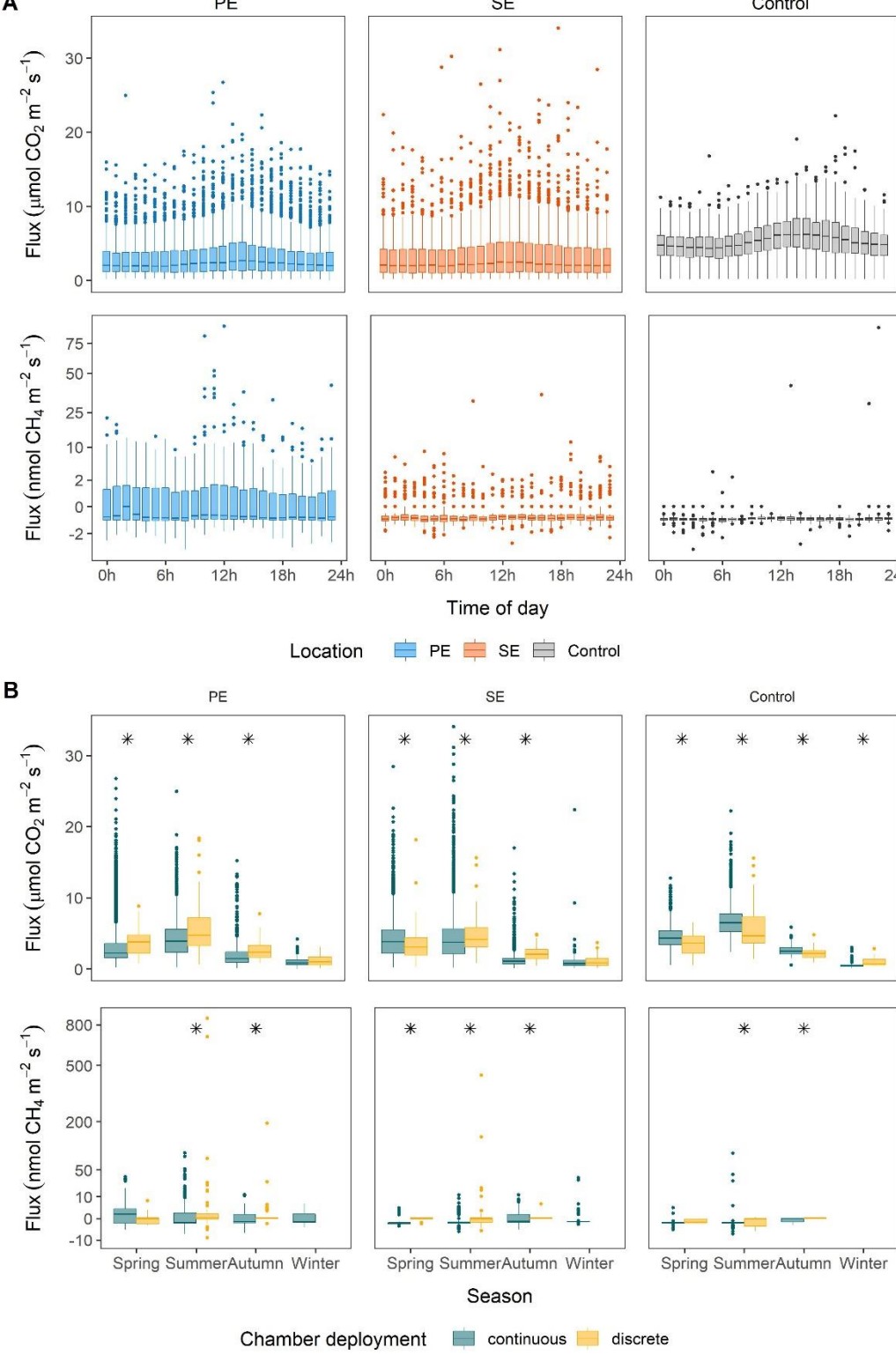

**Figure 4:** (A) Boxplots of diurnal patterns of observed fluxes over trenches receiving PE, trenches receiving SE and undisturbed soil as Control for $CO_2$ and $CH_4$. (B) Boxplots comparing seasonal patterns of observed fluxes from continuous (blue) and discrete (red) measurements over trenches receiving PE, trenches receiving SE and undisturbed soil as Control for $CO_2$ and $CH_4$. Groups marked with an asterisk had significantly different fluxes observed from both measurement approaches (Wilcoxon signed rank test). Note that in both subfigures the y-scale for $CO_2$ is given in units different from units used for $CH_4$ and that the y-axis for $CH_4$ has been square-root transformed to improve data visualisation.

### 3.3.5 Seasonal Patterns

Apart from diurnal variations, GHG fluxes also followed seasonal patterns with fluxes generally higher in warmer summers compared to colder winters. Both sites were located in the west of Ireland where the relatively strong maritime influence leads to cool and windy winters and mostly mild and less windy summers, with similar rainfall quantities across the year. $CO_2$ fluxes exhibited a clear temperature dependence with lower fluxes with narrower ranges during the colder months (from September until February) and higher fluxes with wider ranges over the warmer months. For $CO_2$, seasonal variation was highest for the control soil with median summer fluxes (Md = 6.47 µmol $CO_2$ m$^{-2}$ s$^{-1}$) being approximately 16-times higher than median winter fluxes (Md = 0.40 µmol $CO_2$ m$^{-2}$ s$^{-1}$). Median STU fluxes for both PE and SE trenches only expressed a five-fold increase from 0.81 and 0.72 µmol $CO_2$ m$^{-2}$ s$^{-1}$ to 3.90 and 3..82 µmol $CO_2$ m$^{-2}$ s$^{-1}$ from the lowest to highest flux season, respectively (Figure 4B). During the summer months (June, July and August), the fluxes from the control area exceeded the fluxes from the surface of the STU. Despite the relative difference in the absolute magnitude of the seasonal effect, the relative distribution of fluxes remained mostly constant throughout all seasons and across all types, with approximately 35% to 47% of total emissions originating from the upper quartile of observed fluxes. For $CH_4$, the picture is more complex. While the control soil and trenches receiving PE expressed the highest median $CH_4$ uptake rates in the summer (-0.43 and -0.37 µmol $CH_4$ m$^{-2}$ s$^{-1}$, respectively), STU trenches receiving SE expressed the highest median $CH_4$ uptake rates in spring (-0.50 µmol $CH_4$ m$^{-2}$ s$^{-1}$) and the lowest median $CH_4$ uptake rates in autumn (-0.20 µmol $CH_4$ m$^{-2}$ s$^{-1}$). In comparison, the control soil appeared to be neutral (Md = 0.00 µmol $CH_4$ m$^{-2}$ s$^{-1}$) and the soil over trenches receiving PE a gross emitter of $CH_4$ (Md = 0.40 µmol $CH_4$ m$^{-2}$ s$^{-1}$ in spring) (Figure 4B). Figure 4B also shows a comparison of the fluxes from the discrete versus the continuous measurement above the STUs. Significant differences (indicated by asterisks) are shown in spring summer and autumn with discrete generally higher than the continuous measurements. However, the opposite was found for the control measurements where the continuous were higher than discrete measurements, which perhaps can be attributed to the time of day that discrete measurements were taking place being more aligned with when effluent was being discharged to the STUs.

Notably, during this study in summer 2018 there was an unusually long period of drought which lasted for approximately three months, which caused severe drying of the soil and soil moisture deficit at both sites (Knappe et al. 2020). The extended drought conditions increased daily median $CO_2$ fluxes over all areas. In summer 2018, daily median $CO_2$ fluxes from STU trenches receiving PE, SE and the control soil increased to approximately 450 %, 130 % and 160 %, respectively, in comparison

to the month preceding the drought conditions (Table S6). $CH_4$ flux measurements were not conducted during this period. The increases were significant for all three locations with large effect size for trenches receiving PE (p = <.001; $\bar{r}$= .86 [.85; .86]) and the control (p = <.001; $\bar{r}$= .79 [.65; .74]) and small effect size for trenches receiving SE (p = .03; $\bar{r}$= .29 [.03; .50]). As $CO_2$ emissions from microbial respiration in the soil tend to be governed by soil temperature over a wide range of soil moisture contents but become a function of moisture content as the soil dries out (Smith et al. 2003), the extended dry conditions with SMD of up 75 mm and VWC of <21% as experienced on both sites will lead to increased gas fluxes. The stronger relative flux increases over trenches receiving PE could be related to generally wetter conditions observed at the infiltrative layer as compared to trenches receiving SE (Knappe et al., 2020), thus not leading to moisture-limited gas flux conditions at depth. This becomes evident when comparing flux correlations with environmental factors (Figure 5 and S3). While overall $CO_2$ fluxes were strongly positively correlated with temperature and SMD, and strongly negatively correlated with VWC, fluxes during the drought expressed overall weaker correlations to these environmental factors and were even slightly negative correlated with soil temperature and slightly positive correlated with VWC over trenches receiving PE. This indicates that the presence of sufficient levels of moisture content in the soil became the dominant driver under these conditions.

### 3.3.6 Environmental drivers of GHG fluxes

To assess environmental drivers of observed GHG fluxes, a correlation analysis between fluxes and environmental parameters relating to air and soil temperature, volumetric water content and soil moisture deficit, rainfall and wind speed was performed using Spearman's rank correlation coefficient (Figure 5). Overall, $CO_2$ fluxes showed the most consistent correlations across both control and STU trenches, suggesting that environmental factors were the main driver for controlling $CO_2$ emissions, as opposed to treatment specific differences. $CO_2$ fluxes were strongly positively correlated with both soil and air temperature presumably linked to faster kinetics of microbial activity with increased temperature, albeit with a generally stronger effect over the control (Spearman ρ = 0.75 to 0.77) as compared to STU trenches (Spearman ρ = 0.51 to 0.59). The similarity between the two temperature measurements was to be expected as the soil temperature measurements, which are commonly conducted alongside flux chamber measurements were taken within the first 5 cm in the shallow soil and, thus, should follow a muted, but similar pattern with respect to ambient temperature conditions. Similar in magnitude but opposite in direction were correlations relating to monitored value of VWC (Spearman ρ = -0.65 to -0.53) and calculated value of SMD (Spearman ρ = 0.57 to 0.70), indicating that drier conditions led to higher $CO_2$ fluxes, independent of measuring location. Considering changes in both water content and SMD over the preceding 24 hours, however, it becomes evident that drying events in already relatively dry soil (i.e. when ΔSMD > 0) was correlated to higher $CO_2$ fluxes than water content changes at more saturated conditions (Figure 5). Correlations to wind speed and 3-hour accumulated rainfall prior to measurement remained weak. More unsaturated conditions will enhance oxygen transfer into the soil biofilms, thereby improving externa; mass transfer and the overall kinetics of microbial respiration and conversion of organics to $CO_2$.

For CH$_4$ fluxes, correlations to environmental parameters were generally weaker and often in the opposite direction when compared to CO$_2$ fluxes. Colder temperatures corresponded generally with CH$_4$ uptake and warmer temperatures with CH$_4$ release for control and SE trench fluxes (Spearman $\rho$ = -0.31 to -0.15). The soil's volumetric water content affected STU fluxes more strongly (Spearman $\rho$ = 0.27 to 0.29) than the control (Spearman $\rho$ = 0.13) which also had less overall variability in both
water content and observed fluxes (Figure 5). Considering the changes in SMD, it becomes evident that high CH$_4$ emission events remained largely limited to periods when the soil in the STU starts to dry out from being nearly saturated. Similar patterns were previously reported from gravel-filled soakaways in Ireland (Somlai et al. 2019). While stable dry conditions were generally marked by lower net fluxes and net CH$_4$ uptake over the STU, wetting events only caused considerable CH$_4$ fluxes from trenches receiving PE. The weaker correlation between transitional soil moisture conditions as compared to CO$_2$
fluxes suggests that CH$_4$ fluxes are more susceptible to short-term changes and immediate soil conditions which is in line with observations by Fernández-Baca et al. (2020) who found both STU and control soils to be either weak sinks or weak sources of CH$_4$ except for periods of changing soil moisture conditions where they became net sources over the first 30 min following a simulated rain event. Truhlar et al. (2016) also observed increasing soil CH$_4$ fluxes with increasing VWC attributing it to the fact that with the methanotrophs are aerobic bacteria, whereas methanogens are anaerobic. This also aligns with more general
findings of higher CH$_4$ emissions and uptake rates under changing soil moisture conditions, as well as with increasing temperature (Swenson et al., 2019; Le Mer and Roger, 2001).

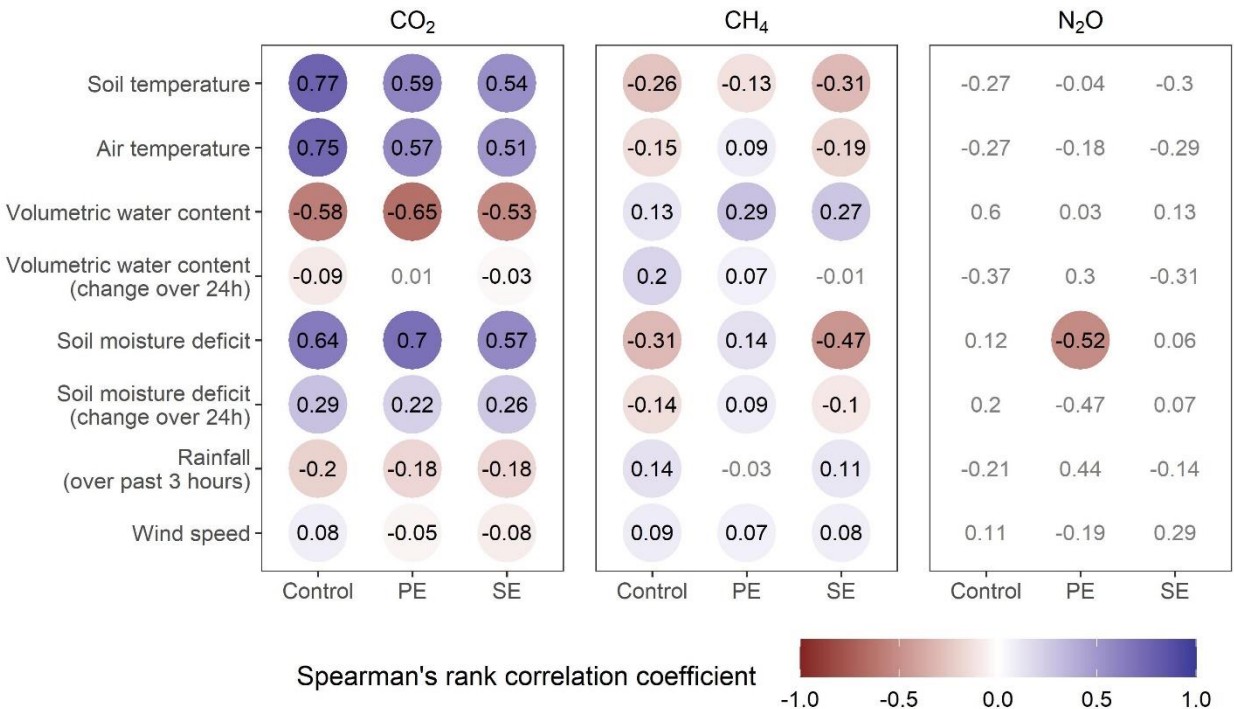

**Figure 5:** Correlograms of observed fluxes over the STU for $CO_2$, $CH_4$ and $N_2O$ to selected environmental parameters. The numerical value and fill color correspond to Spearman's rank correlation coefficient where the correlation was significant (i.e., p ≤ .05). Grey numerical values without colored circles represent correlations that were not significantly different from zero (i.e. p > .05).

Due to the limited amount of available $N_2O$ fluxes which only resulted from discrete measurements, correlations were not significantly different from zero for all parameters and treatment except a moderate negative correlation between $N_2O$ fluxes over trenches receiving PE and the SMD (Spearman ρ = -0.53), however only few measurements occurred at high SMD values, i.e. dry conditions (Figure 5). Truhlar et al. (2016) and Fernández-Baca et al. (2020) found the STU to be net producers of $N_2O$ immediately after rain events; a similar trend observed over trenches receiving PE where 60% of $N_2O$ emission events were recorded after rainfall events within the preceding three hours. Studies in other land use scenarios have also found positive correlations between $N_2O$ emissions and soil moisture (Ambus and Christensen, 1995; Smith et al. 1998), whereby maybe the rapidly changing soil moisture and redox conditions in the soils disrupt the balance of the microbial soil community, leading to periods of partial dentification.

### 3.4 Fluxes from the vents

The STU vent system consists of pipes extending approximately 1 m from the ground surface at the end of percolation trenches. Each vent is directly connected to the perforated effluent distribution pipe within the trench by a 90 degree elbow and capped with a perforated vent cap. Gases originating both within the distribution pipe itself (e.g. from biofilm growth within the pipe itself) and the trench (i.e. through the pipe perforation) can escape through the vent system

For $CO_2$, significantly higher (p = .05) median fluxes from the vent system were observed over trenches receiving PE (Md = 4.91 [2.69; 7.61] µmol $CO_2$ $s^{-1}$) as compared to trenches receiving SE (Md = 2.58 [2.07; 3.88] µmol $CO_2$ $s^{-1}$) with peaks of up to 170.15 and 47.75 µmol $CO_2$ $s^{-1}$, respectively, indicating that a small number of high emission events is responsible for the majority of total observed emissions (Figure 6). The upper quartile of observed fluxes contributed 80.6% and 65.5% of the total recorded emissions, for the PE and SE trenches respectively. Despite generally lower overall fluxes in Site A as compared to Site B, both sites expressed similar patterns between treatments (PE vs. SE; Figure S3). Additionally, fluxes in Site B were marked by a small number of extremely high fluxes (up to 27 times and 18 times higher than median fluxes from trenches receiving PE and SE, respectively). These peaks were generally less pronounced in Site A. The high variability in observed fluxes resulted in a relatively large uncertainty of the overall effect size of this difference between both treatments, ranging from nearly no to moderate effect ($\bar{r}$= .23 [.02; .46]) of pre-treatment level on observed $CO_2$ fluxes.

Similarly, for $CH_4$, significantly higher (p = .003) median fluxes from the vent system were observed over trenches receiving PE (Md = 2.59 [0.23; 11.0] nmol $CH_4$ $s^{-1}$) in comparison to trenches receiving SE (Md = -0.06 [-0.23; 0.05] nmol $CH_4$ $s^{-1}$)

where a small net uptake was observed as a result of lower than ambient $CH_4$ concentrations within the vent system (Figure
6). It is unlikely that this gradient actually led to a passive uptake of $CH_4$ from the natural environment. It is more likely a
result of constant diffusion of $CH_4$, which is less dense than air, thus escaping more readily through the vent system as
compared to $CO_2$, indicating that while the exact long-term $CH_4$ emissions from the vent system might not be effectively
captured by discrete measurements alone, trenches receiving effluent of low organic strength might not be significant producers
of net $CH_4$ emissions. However, peak fluxes of up to 318.0 and 1.20 nmol $CH_4$ s$^{-1}$ for trenches receiving PE and SE,
respectively, suggest that, at least for trenches receiving PE, periodic high emission events determine the majority of observed
total emissions (as discussed previously in Section 3.3.6) with the upper quartile of observed fluxes contributing 85.8% to the
total recorded $CH_4$ emissions. Despite generally lower overall fluxes in Site A as compared to Site B, both sites expressed
similar patterns between treatments (PE vs. SE; Figure S4). The observed effect ($\bar{r}= .46$ [.16; .67]) of pre-treatment on $CH_4$
fluxes from the vent system was stronger than for $CO_2$ fluxes, likely due to the relatively small spread of fluxes observed from
SE trenches.

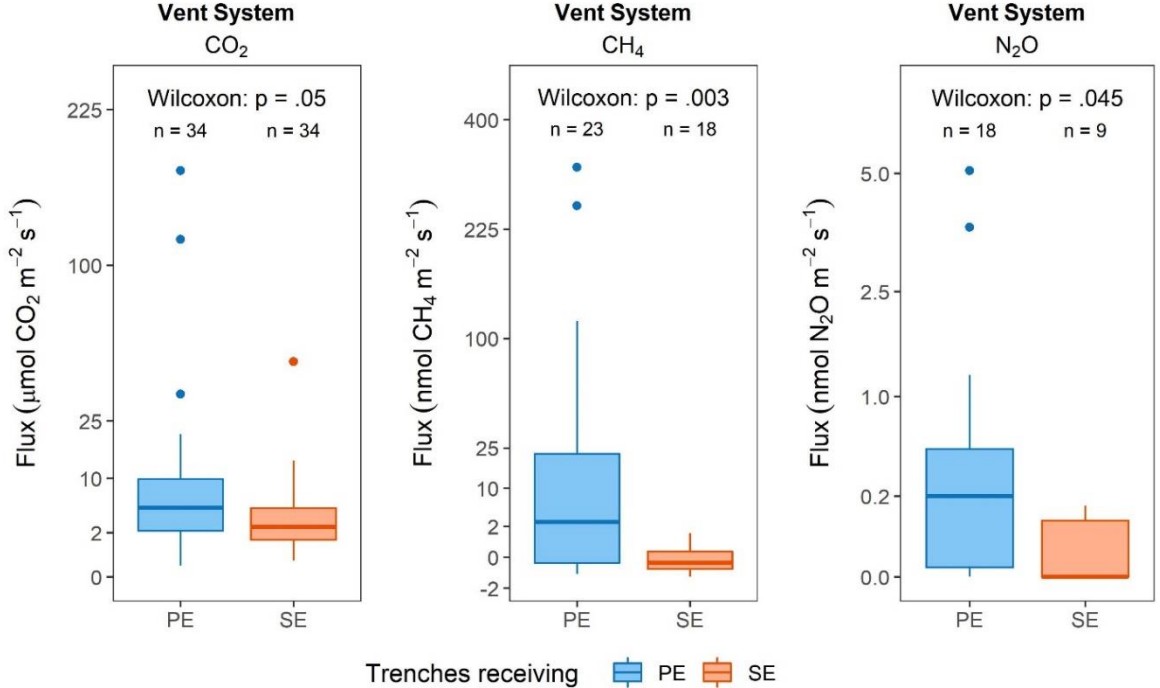

**Figure 6:** Boxplots of observed gas fluxes from the STU vent system for $CO_2$, $CH_4$ and $N_2O$ over trenches receiving primary
(PE) and secondary (SE) effluent. Statistical results are presented as p-value of Wilcoxon signed rank tests with estimated
effect size r and corresponding boot-strapped 95% confidence intervals; n denotes the number of observations per group. Note
that the y-scale for $CO_2$ is given in units different from units used for $CH_4$ and $N_2O$ and that the y-axis has been square-root
transformed to improve data visualisation.

For N$_2$O, significantly higher (p = .045) median fluxes from the vent system were observed over trenches receiving PE (Md = 0.41 [-0.02; 0.49] nmol N$_2$O s$^{-1}$) as compared to trenches receiving SE (Md = 0.101 [0.001; 0.102] nmol N$_2$O s$^{-1}$) (Figure 6). However, the inclusion of zero within the estimated confidence intervals around the median suggest that, despite higher emissions from PE trenches, emissions from both trench types appear to be periodic in nature. Peak N$_2$O fluxes of up to 5.07 and 0.16 nmol N$_2$O s$^{-1}$ were observed from the vents connected to trenches receiving PE and SE, respectively. As with CH$_4$, a large proportion (82.1%) of the total recorded emissions stems from only the upper quartile of observed fluxes from PE trenches. Again, generally lower overall fluxes were recorded in Site A as compared to Site B, but both sites expressed similar patterns between treatments (PE vs. SE; Figure S3). Similar to CO$_2$ and CH$_4$, fluxes in Site B were marked by a small number of extremely high fluxes events (up to 12 times higher than the median fluxes from the PE receiving trenches). These peaks were generally less pronounced in Site A. Increasing the number of measurements in future studies should help elucidate if the difference is indeed significant over the long-term or just an artefact of the limited number of samples collected in this study. The overall effect size ($\bar{r}$= .39 [.07; .65]) of the level of pre-treatment on observed N$_2$O fluxes was small to moderate and similar to the one observed for CH$_4$. Similarly, these findings suggest that while the exact long-term dynamics from the vent system might not be effectively captured by discrete measurements, as trenches receiving effluent low in TN might not be producers of detectable N$_2$O fluxes.

In summary, vent system fluxes of all three gases were generally higher from trenches receiving PE with higher organic and TN load as compared to trenches receiving lower strength SE. This is believed to be a consequence or both the higher substrate availability for microbial degradation processes taking place within the STU trench (thus leading to higher C and N turnover) and the improved effluent dispersal along the trench due to the development of a longer and less permeable biomat at the infiltrative surface of PE trenches (thus extending the microbially active zone further along the trench, i.e. closer to the vent, see Knappe et al. 2020).

These relatively high fluxes coming out of the vents would seem to reinforce the findings on the STUs of higher fluxes picked up over the percolation trenches than between them (see Section 3.4), indicating that the gases are finding an easier escape route to atmosphere via the vents in those regions rather than up through the soil. Indeed, other studies on the sites investigating the transformation of organics in the STUs, showed that the main reduction in organic C occurs just below the infiltrative surface whereby protein-like compounds in the effluent are removed leaving more recalcitrant organics with higher degree of humification and aromaticity to percolate further down through the soil (Dubber et al., 2021) As the vent system is directly connected to the pore space within STU trenches, temperature gradients between the subsurface trench and the atmosphere as well as wind over the vents will cause an upward draw of air from the vent. Fluxes are, thus, expected to be highly variable on timescales of minutes to hours. This corroborates the previous findings that continuous or semi-continuous measurements are much more likely to detect periods of high fluxes (Somlai-Haase et al. 2017; Somlai et al. 2019; Truhlar et al. 2019). Vent flux values obtained from discrete measurements are, thus, challenging to be compared between sites.

The observed vent system fluxes of all three gases were similar to emissions from DWWTS vent systems found in previous studies, e.g. Diaz-Valbuena et al. (2011) reported fluxes of 54.4 µmol $CO_2$ $s^{-1}$ from vents located over the pipe leading from the ST to the STU, which would result in fluxes of 13.7 µmol $CO_2$ $s^{-1}$ per trench if the effluent was split between four STU trenches as in this study. However, the same study found $CH_4$ and $N_2O$ fluxes to be negligible from the same vent.

### 3.5 Total GHG Emissions – comparison between treatment configurations

The total estimated net emissions from the full systems (septic tanks, STUs and vent pipes) have been calculated by assuming that all four trenches receive either only PE or only SE and that the biomat had spread to 15 m in PE trenches and 8.75 m in SE trenches (see Knappe et al., 2020). A comparison has been made at each site to evaluate how the inclusion of up-front packaged secondary treatment units impacts on the net emissions from the STUs (see Table 2). Averaged across both sites, the total net emissions from the STUs (including vent systems) equates to 9.99 kg-$CO_2$eq $cap^{-1}$ $yr^{-1}$ assuming all four trenches

receive PE compared to 0.55 kg-$CO_2$eq $cap^{-1}$ $yr^{-1}$ from the full system assuming all four trenches receive SE. Again, it is important to note here that emission rates were not measured from the packaged secondary treatment units directly and so they have not been included in this calculation. There are tens of different secondary units commercially available on the market, based on different secondary treatment processes (suspended growth systems, biofilm based and hybrid systems) with different hydraulic configurations and so an assessment of this treatment step would be very system-specific and emissions will vary

between the available technology options. It should also be noted that in Site B, the difference between emissions from the STUs receiving PE and SE was significant, whereas in Site A, there was no significant difference between the two total net emissions, reflecting the poorer secondary treatment performance of the up-front packaged treatment plant on Site A.

For the more common septic tank and STU (PE) DWWTS configuration, which accounts for 89% of DWWTSs in Ireland for example, 62.6, 27.5 and 9.9% of the total net emissions were in the form of $CO_2$, $CH_4$ and $N_2O$ (when converted to $CO_2$eq).

**Table 2:** Total net greenhouse gas emissions from the two different treatment configurations: septic tank-STU (PE) and septic tank-packaged secondary treatment system -STU (SE). All emissions are given in kg-$CO_2$eq $cap^{-1}$ $yr^{-1}$.

| | | | STU (surface & vents) | | |
|---|---|---|---|---|---|
| $CO_2$eq emissions | | ST | Surface | Vents | **Total** |
| PE | $CO_2$ | 2.97 | -2.78 | 6.06 | **6.25** |
| | $CH_4$ | 2.72 | 0.00 | 0.03 | **2.75** |
| | $N_2O$ | 0.00 | 0.86 | 0.13 | **0.99** |

| | | | | | |
|---|---|---|---|---|---|
| | total | **5.69** | **-1.92** | **6.22** | **9.99** |
| SE | $CO_2$ | 2.97 | -8.91 | 3.18 | **-2.76** |
| | $CH_4$ | 2.72 | 0.00 | 0.00 | **2.72** |
| | $N_2O$ | 0.00 | 0.57 | 0.02 | **0.59** |
| | total | **5.69** | **-8.33** | **3.20** | **0.55** |

From a treatment train perspective, the total estimated annual $CO_2$ emissions from the STs were 2.97 kg-$CO_2$ cap$^{-1}$ yr$^{-1}$, with higher $CO_2$ emissions in the first chamber at both sites. The total estimated annual $CH_4$ emissions from the STs were 2.72 kg-$CO_2$eq cap$^{-1}$ yr$^{-1}$, with higher $CH_4$ emissions in the second chamber at Site A and higher $CH_4$ emissions in the first chamber at Site B. As stated previously, these emissions from the ST may be biased towards underestimating the actual emissions as those were mainly based on discrete measurements.

From the surface of the STUs, the total annual $CO_2$eq emissions estimated for the areas receiving PE were -1.92 kg-$CO_2$eq cap$^{-1}$ yr$^{-1}$, i.e. slightly less compared to emissions from the adjacent native soil. However, this is thought to be mainly caused by the percolation trenches acting to channel some of the microbially produced gases within the STUs out to the vents, which equated to total annual $CO_2$eq emissions of 6.22 kg-$CO_2$eq cap$^{-1}$ yr$^{-1}$, primarily in the form of $CO_2$. In the more lightly organically loaded STUs receiving SE there was a much greater defect between the emissions from the surface of the STU compared to the adjacent undisturbed soil, equating to-8.33 kg-$CO_2$eq cap$^{-1}$ yr$^{-1}$. When the total annual $CO_2$eq emissions from the vent systems receiving SE of 3.20 kg-$CO_2$eq cap$^{-1}$ yr$^{-1}$, are taken into account, this still shows overall lower emissions equivalent to -5.14 kg-$CO_2$eq cap$^{-1}$ yr$^{-1}$ compared to the native soil, which suggests that either lower the strength organic effluent entering the STU is leading to a change in microbial diversity within the soil and thereby different net emissions and/or the gases are finding an alternative pathway to the atmosphere, possibly back up via the distribution boxes at the head of the trenches as the active biomat was shown to extend only for the first few meters in these SE-fed trenches. As mentioned, $CO_2$ emissions were not measured from the secondary treatment plants themselves due to access restrictions into the modules, but presumably these would be significant due to the reduction in organics in the effluent that occurs within them. The impact of the difference in performance between the different packaged secondary treatment units on the two sites was clearly picked up with higher emissions from trenches receiving SE in Site A and slightly higher emissions from trenches receiving PE in Site B.

A comparison of the net GHG emissions from the septic tank and STU (9.99 $CO_2$ kg-$CO_2$eq cap$^{-1}$ yr$^{-1}$) against other similar studies of on-site wastewater treatment reveals much lower fluxes. Diaz-Valbuena et al (2011) estimated total GHG emissions

from septic tanks and vent systems (but did not measure fluxes directly above the STU) of 101 to 119 kg-CO$_2$eq cap$^{-1}$ yr$^{-1}$, whilst Truhlar et al (2016) estimated even higher total GHG emissions of 270 kg-CO$_2$eq cap$^{-1}$ yr$^{-1}$ (with 50 kg-CO$_2$eq cap$^{-1}$ yr$^{-1}$ from the STU). A life cycle assessment of nearly 800 septic tank systems without soil dispersal in Poland showed that approximately 27% of total cradle-to-grave GHG emissions (i.e. 5.21 kg CO$_{2\,eq}$ cap$^{-1}$ yr$^{-1}$) were released during the operational phase of the system, of which about half was attributed to direct GHG emissions to the atmosphere (Burchart-Korol et al., 2019). Whilst it is difficult to generalize about GHG emissions associated with centralized wastewater treatment systems used to serve urban populations given the different number of permutations (and studies) on GHG emissions from different process combinations and size of plants, the Ecoinvent database used for Life Cycle Analysis Ecoinvent (2021) returns similar GHG emissions of 346 and 349 kg-CO$_2$(eq) cap$^{-1}$ yr$^{-1}$ for population equivalents of approximately 3,000 and 200,000, respectively, which provide an interesting comparison to the on-site wastewater treatment systems.

Finally, this study provides data that can be used to refine emission factors used for onsite wastewater treatment in terms of IPCC national accounting for CH$_4$ and N$_2$O emissions (note, CO$_2$ emissions are not counted in such accounting as these are considered to be of biogenic origin). In the recent 2019 Refinement to the 2006 IPCC Guidelines for National Greenhouse Gas Inventories, on-site systems are considered under the uncollected wastewater category, and have been broken down into septic tank systems with and without dispersion fields (i.e., STUs); there are no further categories of different permutations of on-site system design, as have been evaluated in this study. The IPCC emission factors for septic tank systems with dispersion fields are stated at 0.125 kg-CH$_4$/kg-COD and 0.0045 kg-N$_2$O-N/kg-N, which can be compared against the mean emission factors from this research as 0.0036 kg-CH$_4$/kg-COD and 0.0003 kg-N$_2$O-N/kg N respectively, an order of magnitude lower. This would therefore suggest that the IPCC guidelines are significantly over-estimating GHG emissions associated with on-site wastewater treatment at present. The guidelines do presume that all CH$_4$ emissions are produced within the septic tank (with negligible emission from the STU), whilst N$_2$O emission are produced in the effluent dispersal system in the soil, which would seem to be corroborated by this study. Using improved emission factors based on recent studies, a survey of the global impact of decentralised sanitation technologies (Shaw et al., 2021) found that septic tank systems were amongst the main contributors of sanitation related GHG emissions in all scenarios that would allow to achieve Sustainable Development Goal (SDG) 6.2 (i.e., ending open defecation by 2030). Although total sanitation-related emissions are expected to increase, they are comparatively very small (0.2% of total global anthropogenic CO$_2$ emissions) and offer affordable and scalable forms of sanitation which yield the potential for energy capture and reuse (i.e., lower emission potential) in the form of CH$_4$ capture directly at source.

### 3.6 Minimum data set requirement to gain accurate flux estimates

Finally, an attempt has been made to determine a minimum data set (spatially and temporally) required using the more commonly available discrete measurement methodology which might approximate the accuracy of flux estimates made using

continuous measurements. Overall, observed soil GHG fluxes showed clear diurnal and seasonal patterns and were correlated to environmental parameters such as temperature and moisture status in the soil. Continuous measurements appeared to provide a more accurate representation of the full variability of fluxes as survey measurements are usually limited to day-time hours. In practice it might, however, not always be possible to deploy long-term continuous measurements in the field. A linear mixed effects (LME) model similar to Truhlar et al. (2019) for $CO_2$ and $CH_4$ fluxes was used to estimate the effect of different spatially-distributed discrete sampling techniques on regression coefficients. The LME model included site (Site A, Site B) and location (PE, SE, Control) as random effects and the four highest correlated environmental factors (soil and air temperature, VWC, SMD) as fixed effects. As the limiting factor for discrete sampling will most likely be manual work by field personnel, we focused on two realistic sampling strategies: i) fewer measurements per day but measurement over several consecutive days (i.e. one morning and one evening measurement per day on each weekday), and ii) more measurements on a single day but fewer overall sampling days (i.e. eight daytime measurements per day in two-hour intervals during daytime hours on one day per week). Comparing the distribution of modeled parameter estimates (Figure S5) with the outcomes from models using the full dataset from continuous and discrete measurements, respectively, we conclude that while neither discrete approaches can replicate the results obtained from the continuous measurements, the approach of sampling more frequently during the day on fewer overall days provided regression coefficient point estimates closer to these of the full dataset (i.e. higher accuracy) as compared to sparser sampling on consecutive days. Resulting coefficient estimate distributions however were comparatively wide (i.e. low accuracy) for both $CO_2$ and $CH_4$, but generally better for $CO_2$. As shown in this and other studies, microbially-induced gas fluxes are generally driven by C-availability in the soil, environmental factors and soil characteristics, it is important to sample over a preferably wide range of temperature and moisture conditions to optimally capture natural and system-specific gas flux variations.

## 4 Conclusions

This study provides the first field-scale comparison of the effect of different levels pre-treatment on greenhouse gas emissions from on-site wastewater treatment systems across a year. This research demonstrates that GHG emissions from the different parts of the DWWTSs are highly variable and correlated to environmental factors and water usage patterns. The highest measured $CO_2$ flux rates were observed from the STUs at both sites; however, when these rates were adjusted to account for the background soil emissions, one STU was a relatively high net emitter of $CO_2$ emissions compared to the other STU which was apparently emitting less $CO_2$ emissions in comparison to the background soil. However, both sites were also characterised by high emissions from their vent systems for the percolation pipes, implying that much of the gases generated by microbial soil treatment processes on the percolating effluent were escaping via the vent system as compared to the soil. Vent fluxes were characterised by a low number of high emission events which were responsible for the majority of total observed vent emissions, indicating that improved measurement techniques would be needed to accurately assess vent emissions in the future.

The STs contributed to the highest $CH_4$ emissions at both sites, as found in other similar recent studies (Diaz-Valbuena et al. 2011 etc.) and the highest $N_2O$ fluxes were measured in the vent systems at both sites.

The total net GHG emissions from a conventional septic tank system with STU base on the results from both sites is 9.99 kg-$CO_2$eq yr$^{-1}$ cap$^{-1}$. Approximately 63% of the total net emissions was in the form of $CO_2$, around 27% in $CH_4$ and less than 10% in $N_2O$. Comparing a hypothetical on-site treatment system with and without packaged secondary module equated to an additional 9.44 kg-$CO_2$eq cap$^{-1}$ yr$^{-1}$ of emissions from the STU system if it receives PE directly as compared to SE from the secondary treatment module, with much of that difference probably emitted by the secondary treatment package system.This study therefore has provided insights into implications for managing GHG emissions from DWWTSs that can be attained by different system configurations as well as providing data that suggests that the current IPCC emission factors for $CH_4$ and $N_2O$ are significantly overestimating emissions for standard on-site wastewater treatment systems comprising of a septic tank and soil treatment area.

**Acknowledgements**

This work was supported by the Science Foundation Ireland under grant number 13/IA/1923. The authors thank the house owners for enabling continued access to the research sites, as well as Patrick Veale for assistance with field work.

**Code/Data availability**

The datasets that this research has been based on can be found in Knappe et al., (2022), "Data for: Assessing the spatial and temporal variability of GHG emissions from different configurations of on-site wastewater treatment system using discrete and continuous gas flux measurement", *Mendeley Data, V1*, doi: 10.17632/s6jmnhmyzf.1.

**Author contributions**

L.G. conceived the study and gained the required funding. L.G., J.K. and C.S. planned the experiments. J.K. and C.S. performed the experiments and analysed data. All authors interpreted the results, wrote and reviewed the manuscript.

**Competing interests**

The authors declare that they have no known competing financial interests or personal relationships that could have appeared to influence the work reported in this paper.

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
