# Peer review of "Assessing the spatial and temporal variability of GHG emissions from different configurations of on-site wastewater treatment system using discrete and continuous gas flux measurement"

_Biogeosciences, 2021_

## Author Response (AR1)

**Response to Reviewer 1**

The manuscript describes greenhouse gas flux measurements from a constructed domestic waste water treatment system (using percolation through soil along trenches). The study reasonably includes all stages of effluent, from septic tanks to trenches (where effluent is introduced below-ground through pipes), and vents to the atmosphere. Collecting this kind of data is challenging, given the seasonal and diurnal fluctuations, and events such as ebullition and flushing through vents. The combination of spot measurements and use of continuous chambers is hence appropriate.

We thank the reviewer for their time and thoughtful comments to improve this manuscript.

My main issue with the analysis is the framing of fluxes in comparison to control soils. This may seem reasonable, as any changes from undisturbed ("natural") conditions would be attributed to the use of effluent treatment. However, the results indicate significantly reduced fluxes of $CO_2$, which the authors interpret as a net sink of $CO_2$ through effluent treatment. It is grossly simplistic, as the fate of C introduced from domestic effluent is not explained or even considered. To become a sink of $CO_2$, the soils have to assimilate C somehow, but no such mechanism is proposed. Measurements use dark chambers, so potential fertilisation effects on vegetation resulting in increased photosynthesis can not account for this finding. It is more likely that the soil disturbance, including replacement of soil by gravel, has reduced $CO_2$ flux artificially in the treatment beds. A true control treatment would have been required, with trench construction and hence disturbance identical to that of the effluent treatment beds.

We acknowledge the comment regarding the lack of a "true control" raised by the reviewer. As the reviewer has stated, it is very challenging to collect such kind of data and we present here the first data set of this kind, including long-term and discrete measurements from both septic tanks and soil treatment units. As the study sites were built before this study commenced, we were unfortunately not able to include the construction of this "true control", i.e. gravel trenches not receiving any effluent. However, we do clearly acknowledge the value of such a control measurement for future studies and will include this into the overall discussion of the results. The framing of observed fluxes in comparison to control soils will be revised accordingly and a clearer distinction between the soil treatment unit fluxes and natural soil fluxes will be included in the revised manuscript. For the time being, we understand that the control as conceptualised in this this study, might not be a "true control" but rather a "best available control" in the field.

The reviewer makes an interesting point with respect to the impact of soil disturbance that was needed to install the percolation trenches and gravel and its potential impact on the flux differences between the undisturbed soil control site. We agree that this may make some difference, but it should be noted that most of the soil removed for the trenches was the subsoil (with very little organic matter in it) and that once the trenches were installed, the more organic rich topsoil (where presumably most of the natural control carbon cycling from the vegetative grass layer is occurring) was replaced. Of course, this soil had been disturbed and also some soil has been removed where the gravel now sits and so this may account for some of the differences highlighted by the reviewer, but the method of construction does need to be considered in the comparison. Another point is that the width of the trenches is only 0.5 m wide which corresponds to a relatively minor fraction of area of

the STU from which soil was excavated. As can be seen from Figure 1 the gas sampling was randomly spread out across the whole STU known to be receiving effluent and so many of the chambers were sitting on undisturbed soil profiles (similar to the control). In Section 3.3.2 we show that the net $CO_2$ fluxes were measured from positions located between trenches were higher compared to inserts located above trenches, which indicates that the gases being generated in and below the trenches are making their way to the atmosphere via the percolation pipes via the vents (and not up through the soil). However, for the SE fed trenches, even when the emissions from the vents are taken into account, the combined emissions from the STU (surface and vents) are still lower compared to the native soil, which suggests that either lower the strength organic effluent entering the STU is leading to a change in microbial diversity within the soil and thereby different net emissions and/or the gases are finding an alternative pathway to the atmosphere, possibly back up via the distribution boxes at the head of the trenches as the active biomat was shown to extend only for the first few meters in these SE-fed trenches.These are all interesting points which need to be brought out more clearly in the discussion of the revised paper.

The fate of C in the soil is briefly discussed in the current version of the manuscript (L399-406 + Conclusion section) as previous indicated the presence of methanogens in the top layer of soil above treatment trenches. However, we acknowledge that we should have included more discussion of our results in relation to the potential C & N processes and pathways in the soil in light of the findings that we are presenting. Parallel research on the characterisation of organic matter and its transformation processes in on-site wastewater effluent that we have been carrying out which has just been published (Dubber et al., 2021) has clearly shown that most of the breakdown of protein-like organic compounds in the effluent is broken down (leaving more humified organics) in the biomat at the infiltrating layer in the trenches. Hence, the $CO_2$ and $CH_4$ produced due to this decomposition is most likely to move into the gravel and back into the free space of the percolation pipes and so will leave the STU via the vent pipes (as opposed to up though the soil) – as per the previous point made about the positions of the monitoring points. Any wind dragging on the top of these pipes would cause a slight negative pressure and so encourage gasses from the soil that accumulate in the pipes to flow out via the vents. We have quantified these emissions in this paper which equate to the highest fluxes in both STE and SE systems and so these can be considered to be where most of the actual emissions from the breakdown of the organics in the effluent from STU end up. This point has now been made more clear in the revised manuscript. In addition, it should be noted that we are currently carrying out a follow-on research on the sites to investigate the microbial diversity at different depths within the soil in the soil treatment units which is providing additional insights into the relative abundance of different microorganisms present (and hence biogeochemical processes linked to the GHG emissions).

In the revised manuscript we have also adjusted the phrasing of fluxes as net fluxes which should hopefully make the interpretation of the results clearer for a reader.

To be publishable, the authors have to re-evaluate the flux calculation (which is never presented or explained in the methods). The assumed "net $CO_2$ sink" has a major influence on the net GHG balance, which is not robust, and can not be presented as such.

The flux calculations used in this paper are described in detail in the Supplemental Information and have previously been published in Somlai-Haase et al. (2017), as referenced in L96 in the Method section. However, the revised manuscript now includes this

detail in the Supplemental Information which is now referenced. We will also make more clear the difference between the actual C-fluxes to atmosphere from these sites and then what is meant by net C-emissions (notwithstanding the previous points that have been made about the exact nature of the "control" used in this study).

There is relatively little discussion of results. References are included to compare individual aspects to literature, but I missed a comprehensive evaluation. Maybe a separation of 'Results' and 'Discussion' would work better.

As this is a rather new field of study, there appear to be very few published manuscripts on GHG emissions from such systems against which we can relate and compare our results. However, we will adjust the discussion of our results to better reflect the current available literature (including more discussion of the actual processes that might be leading to the different fluxes) in order to provide a more comprehensive evaluation. Six new references have been added to the manuscript (Burchart-Korol et al., 2019; Dubber et al., 2021; Huynh et al., 2021; Mac Mahon et al., 2022; Shaw et al., 2021; Smith et al, 2003).

On balance, I struggle to justify the publication in Biogeosciences. Microbial processing and GHG implications of domestic effluent is included, which is good, but I missed a reliable treatment of interaction with soil biota, and consideration of sources and sinks beyond a simplistic flux comparison (which would make this paper relevant to BGS). It's a shame, as the data set is impressive, and surely useful for assessing GHG impacts such domestic schemes. Unfortunately, the designed is flawed by lack of a true control, making it difficult to provide robust GHG budgets.

The reason that we choose Biogeosciences as a journal to publish our work in is that the journal's stated aim is "*dedicated to the publication and discussion of research … on all aspects of the interactions between the biological, chemical, and physical processes in terrestrial or extraterrestrial life with the geosphere, hydrosphere, and atmosphere. The objective of the journal is to cut across the boundaries of established sciences and achieve an interdisciplinary view of these interactions*. We feel that our study does match these aims as it is studying with the effluent percolating through the soil (i.e., the geosphere and hydrosphere) and is quantifying the production of GHGs to the atmosphere. The contaminant attenuation of effluent as it percolates through soil involves many different microbiological, chemical and physical processes (as has been investigated in research studies by us and others into such on-site treatment processes and discussed with appropriate references in the Introduction section). However, rereading our paper now, the reviewer is correct that we do not elucidate very much on how these processes are acting to produce the fluxes of GHGs that we are presenting in this paper, which we will address in a revision.

As mentioned before, previous studies have linked the presence of methanogens in the top layer of soil above treatment trenches to reduced soil fluxes as compared to expected values when taking direct in-trench/vent system measurements into account. A more detailed discussion of our results in light of these findings will hopefully alleviate the reviewers concerns and provide a reliable discussion of interaction with soil biota.

We have provided a previous response to the issue about a true control, which we agree does need some more consideration and discussion in the paper, but this does not negate

all of the actual fluxes that were measured (both spatially and temporally) for the three different GHGs at the different locations (in the septic tank, above the soil treatment and from the vent pipes) which have all been quantified in what we feel is the most comprehensive study yet to be published (from what we can see in the current literature). Hence, we feel the paper contains very valuable research findings and conclusions for the scientific and wider community.

Detailed comments

76: Can you give at least some detail of what a 'rotating biological contactor' is?

The revised manuscript will include a short description of an RBC for readers not currently familiar with its basic design principles. It is a fixed film secondary wastewater treatment process in which plastic discs slowly rotate bring the attached biofilm down into the sewage (the substrate) and then up into the air (for oxygen transfer). They are usually arranged in two separate sequential pockets for such small-scale on-site packaged systems with the first chamber usually achieving removal of biodegradable organics via heterotrophic bacteria, whereas the second chamber usually then nitrifies the effluent (i.e., ammonia to nitrate)  via autotropic bacteria

103: I'm unsure what "each of the two ST chambers" means. There seems to be no descriptions of "chambers" of STs, and it becomes confusing when you describe flux chamber measurements.

The current version of the manuscript is indeed lacking a description of the ST as a two chamber system. The revised manuscript will include a clarification in the Research Sites section.

167-184: Section 3.2 reports results that are not part of this study, and don't relate to methods presented earlier. Please either present methods of measurement to obtain these values in the methods, or integrate the information provided here into the description of sites and STUs.

The results from water quality measurements are summarized from Knappe et al. (2020) and provided here for context. We understand that it might improve the manuscript by, instead of mentioning these results in the Results section, referring to the relevant information once the results of the flux measurements are discussed and will adjust the manuscript accordingly. Hence, Section 3,2 has now been deleted and most of the text now moved to Section 2.3.

232/233: This should be Figure 2B (?)

Yes, indeed. Thank you for noticing.

269/270: Reported fluxes of 0.00, and a range of [0.00; 0.00] are not meaningful. If fluxes were measured, they should be reported with three significant figures, whatever the magnitude. The same applies in line 274.

Agreed, the revised manuscript will include the results reported with meaningful significant figures.

284: You state "clearly seem to increase", but later state that results are not statistically significant, which is contradictory.

While the linear regressions of both treatments were not significantly different from zero on an individual basis, the phrasing "clearly seem to indicate" refers to a comparison of both regression results to each other. We will make sure that the phrasing in the revised manuscript will make this separation clearer and will provide the results of statistical analysis for the latter separately.

311-314: This largely repeats information already given in the Methods.

Agreed, the repeated information has been deleted in the revised manuscript.

325: If the net flux of CH4 is negative, the STU as a whole (comprising effluent for STs, soil and vegetation) is a net sink of CH4. The 'natural' CH4 sink (measured on control soils) has been diminished due to gross CH4 emissions increasing, but the net term remains negative.

Yes, we can see how this is confusing. In the revised manuscript we will adjust the phrasing of how we report net fluxes as measured by the chambers and relative fluxes where the STU is compared to the fluxes from the native soil, which this should hopefully make such statement clearer.

340: The 'median net uptake' is not well explained, and I think even misleading. The data show positive $CO_2$ fluxes for all treatments, and as I assume the chamber was obscure, there is no realistic prospect of $CO_2$ uptake (but possibly for methane). Here. You seem to refer to the difference between effluent treatments in the STU to control soils, where a lower flux in treatments is deemed a 'net uptake'. If at all, this would be a gross uptake, as the net effect of all processes is evidently still a source of $CO_2$. I suggest moving away from the terminology of 'net uptake' when discussing these fluxes, and provide interpretation of possible mechanisms and processes in the discussion.

This is a good point and in line with the previous comments on "true controls". We will adjust our terminology accordingly and only use net fluxes to describe the results from the chamber measurements (not the relative balance when compared against the controls). Also, as per our previous response, provide more discussion of the possible biogeochemical mechanisms and processes leading to these fluxes.

Figure 4: Extremely high $CO_2$ fluxes are not really plausible (especially in control plots). Have flux regressions been quality-checked?

Yes, flux calculations have been quality-checked as described in Somlai-Haase et al. (2017) and fluxes with unacceptable regression quality parameters were excluded from analysis. A brief overview of the fraction of excluded measurements will be included in Section 3.1 in the revised manuscript.

Figure 4: Colours mean different things between panels, which is confusing. For A, panels already separate location, so no need to use different colours. I assume that the locations are the same in Panel B (i.e. PE on left, Se in middle, Control on right)?

The revised manuscript will include a clearer color pattern in Fig. 4.

353: "in the west of Ireland", or "in western Ireland".

This will be corrected.

354-356: Was there a notable flux response to the drought period, and did it differ between STU and control plots?

Yes, there was a notable response to the drought, particularly for $CO_2$ fluxes (see Figure 5 which shows the correlation of soil moisture vs $CO_2$ fluxes) and the revised manuscript will present a more specific evaluation of this period.

362- 365: This sentence on relative contributions is hard to interpret. Can you explain your distinction of absolute vs. relative fluxes better?

The absolute fluxes are the respective observed flux values as measured by the chambers. Relative fluxes are expressed in relation to control fluxes and would be unitless. We will improve the terminology here.

424-427: Avoid repeating methods here.

The repeated text has now been deleted.

459: The same problem with presenting significant figures. Please don't present fixed number of decimal places, but instead always the same number of significant figures for all fluxes.

This will be corrected.

479: "higher lower"?

Higher

509/510: This is not correct. Units should be kg $CO_2$-eq cap-1 yr-1. Likewise in lines 622 and 623. (Please check throughout text!)

Well spotted. Thank you. These have been corrected.

Figure S2: Please provide more information in the figure caption to allow readers to follow what's shown without reading the text in detail. What are the two colours, and what are "chamber 1" and "chamber 2"?

This has been be corrected.

**Response to Reviewer 2**

This manuscript describes differences in greenhouse gas fluxes measured continuously or discretely from two onsite wastewater treatment systems that include secondary treatment as part of the treatment train: one with a rotating biological contactor, the other with a coconut husk media filter. The treated water is dispersed to a soil treatment unit and, in both cases, untreated septic tank effluent is also dispersed to the STU. Comparisons of flux values obtained using continuous and discrete measurements are made for the septic tank, the soil above the STU, and the vents at the end of the pipes that deliver effluent to the STU. GHG fluxes from the STU are compared to those from a Control area.

We thank the reviewer for their time and thoughtful comments to improve this manuscript.

There are a number of issues that I think need to be addressed:

1. The difference in $CO_2$ flux between Control and STUs is often negative, that is, the STU is somehow acting as a sink for $CO_2$. The possible mechanism(s) by which this takes place are not really discussed in the paper. Very few microbial processes assimilate $CO_2$ in wastewater (e.g., autotrophic ammonia oxidation), and these would likely be minimized by both secondary treatment processes, which promote ammonia oxidation before it reaches the STU. One large difference between the Control and STU soils is the absence of subsurface horizons in the latter, which would have been removed to install the effluent delivery system. The removed soil would contribute to $CO_2$ flux at the soil surface which, when compared to Control soil, would have a lower $CO_2$ The authors should, then, reconsider comparisons with Control soil, not only for $CO_2$, but for all three gases (assuming they don't have data for an STU that did not receive effluent), since gross consumption and production of $CH_4$ and $N_2O$ can take place in the "missing" soil.

   We acknowledge the comment regarding the lack of a "true control" raised by the reviewer. As the study sites were built before this study commenced, we were unfortunately not able to include the construction of this "true control", i.e. gravel trenches not receiving any effluent. However, we do clearly acknowledge the value of such a control measurement for future studies and will include this into the overall discussion of the results and Conclusions section of the manuscript. The framing of observed fluxes in comparison to control soils will be revised accordingly and a clearer distinction between the soil treatment unit fluxes and natural soil fluxes will be included in the revised manuscript. For the time being, we understand that the control as conceptualised in this this study, might not be a "true control" but rather a "best available control" in the field.

   The reviewer makes a valid point with respect to the potential difference between the undisturbed soil control site and the area above the STU in that soil needed to be removed to install the gravel and trenches. This may make some difference, but it should be noted that the main soil removed for the trenches was the subsoil (with very little organic matter in it) and that once the trenches were installed, the more organic rich topsoil (where presumably most of the natural control carbon cycling from the vegetative grass layer is occurring) was replaced. Of course, this soil had been disturbed and also some soil has been removed where the gravel now sits, so this may account for some of the differences highlighted by the reviewer, but the

method of construction does need to be considered in the comparison. Another point is that the width of the trenches is only 0.5 m wide which corresponds to a relatively minor fraction of area of the STU from which soil was excavated. As can be seen from Figure 1 the gas sampling was randomly spread out across the whole STU know to be receiving effluent and so several of the chambers were sitting on undisturbed soil profiles (similar to the control). In Section 3.3.2 we show that the net $CO_2$ fluxes were measured from positions located between trenches were higher compared to inserts located above trenches, which indicates that the gases being generated in and below the trenches are making their way to the atmosphere via the percolation pipes via the vents (and not up through the soil). However, for the SE fed trenches, even when the emissions from the vents are taken into account, the combined emissions from the STU (surface and vents) are still lower compared to the native soil, which suggests that either lower the strength organic effluent entering the STU is leading to a change in microbial diversity within the soil and thereby different net emissions and/or the gases are finding an alternative pathway to the atmosphere, possibly back up via the distribution boxes at the head of the trenches as the active biomat was shown to extend only for the first few meters in these SE-fed trenches.

We acknowledge that we should have included more discussion of our results in relation to the potential C & N processes and pathways in the soil in light of the findings that we are presenting. Parallel research on the characterisation of organic matter and its transformation processes in on-site wastewater effluent that we have been carrying out which has just been published (Dubber et al., 2021) has clearly shown that most of the breakdown of protein-like organic compounds in the effluent is broken down (leaving more humified organics) in the biomat at the infiltrating layer in the trenches. Hence, the $CO_2$ and $CH_4$ produced due to this decomposition is most likely to move into the gravel and back into the free space of the percolation pipes and so will leave the STU via the vent pipes (as opposed to up though the soil) – as per the previous point made about the positions of the monitoring points. Any wind dragging on the top of these pipes would cause a slight negative pressure and so encourage gasses from the soil that accumulate in the pipes to flow out via the vents. We have quantified these emissions in this paper which equate to the highest fluxes in both STE and SE systems and so these can be considered to be where most of the actual emissions from the breakdown of the organics in the effluent from STU end up. This point has now been made more clear in the revised manuscript. In addition, it should be noted that we are currently carrying out a follow-on research on the sites to investigate the microbial diversity at different depths within the soil in the soil treatment units which is providing additional insights into the relative abundance of different microorganisms present (and hence biogeochemical processes linked to the GHG emissions).These are all interesting points which need to be brought out more clearly in the discussion of the revised paper.

In the revised manuscript we will also adjust the phrasing of fluxes as net fluxes which should hopefully make the interpretation of the results clearer for a reader.

2. There are several published studies on GHG emissions from secondary treatment units that show that these can be considerable. The treatment units used in this study both rely heavily on microbial processes to remove and transform C and N, which produces $CO_2$ and $N_2O$. In addition, mechanical mixing and/or turbulent flow in these units tends to result in loss of $CH_4$ and $N_2O$ form effluent to the atmosphere. In the

absence of values for these emissions, the flux values that were measured lack context. Differences in flux between secondary treated effluent and tank effluent could help provide some context.

Due to design and access limitations, we were not able to use the current sampling methodology to assess GHG emission directly from the secondary units. We understand, that this limits the overall applicability of the results for system-wide emissions. However, this manuscript presents the first data set of this spatial and temporal scale for on-site wastewater treatment systems, including long-term and discrete measurements from both septic tanks and soil treatment units. As there are tens of different secondary units commercially available on the market, an assessment of this treatment step would be very system-specific and emissions will probably vary significantly between the available technology options. This point has been added to the text. We thus, limit this study to the parts of the treatment train (septic tank for primary treatment and soil treatment unit for effluent dispersal) that are most likely present in a majority of on-site systems. In Ireland for example, septic tanks with percolation trenches account for an estimated 89% of all on-site systems. In the revised manuscript we will, however, strive to include a selection of results regarding GHG emissions from said studies in our Discussion to help contextualize the result of our study.

3. There is, in general, very little discussion of biogeochemical processes that could explain results in this paper, and limited discussion of results in the context of the current published literature. For the most part flux values are reported and compared within the study, without getting into the biogeochemical and/or abiotic processes that may that drive these in the soil or the effluent. It may be that Biogeosciences is not a good match for this work.

The reason that we choose Biogeosciences as a journal to publish our work in is that the journal's stated aim is "*dedicated to the publication and discussion of research … on all aspects of the interactions between the biological, chemical, and physical processes in terrestrial or extraterrestrial life with the geosphere, hydrosphere, and atmosphere. The objective of the journal is to cut across the boundaries of established sciences and achieve an interdisciplinary view of these interactions*". We feel that our study does match these aims as it is studying with the effluent percolating through the soil (i.e., the geosphere and hydrosphere) and is quantifying the production of GHGs to the atmosphere. The contaminant attenuation of effluent as it percolates through soil involves many different microbiological, chemical and physical processes (as has been investigated in research studies by us and others into such on-site treatment processes and discussed with appropriate references in the Introduction section). However, rereading our paper now, the reviewer is correct that we do not elucidate very much on how these processes are acting to produce the fluxes of GHGs that we are presenting in this paper, which we will address in a revision.

The fate of C in the soil is briefly discussed in the current version of the manuscript (L399-406 + Conclusion section) as previous indicated the presence of methanogens in the top layer of soil above treatment trenches. However, we will adjust the discussion of our results to better reflect the potential C & N processes and pathways in the soil in light of these studies. It should be noted that we are currently carrying

out studies on the sites to investigate the microbial diversity at different depths within the soil in the soil treatment units which is providing additional insights into the relative abundance of different microorganisms present. We are particularly interested in those organisms which are key with respect to the production of GHGs (e.g. the location of methanotrophs versus methanogens, nitrifiers vs denitrifiers, including annamox bacteria etc.). However, this sort of in-depth study will need to be the subject of a different paper in the future as it would be too much to include in this already very long paper.

4. Most researchers working in this area will not have access to the equipment needed for continuous measurements of GHG fluxes; rather, discreet flux measurements are more likely to be made by most. As such, the results of this study could be made more useful by developing a minimum data set (spatially and temporally) required to approximate the accuracy of flux estimates made using continuous measurements. Although I understand this has clear limitations related to climate, treatment train, etc., it would be a good start, and a meaningful contribution to the field.

Yes, this is a very good suggestion. We are planning to make the full data set available under Open Access licensing upon publication of the manuscript, thus enabling other researchers to identify such minimum data sets best suitable for their respective site specification based on our data. However, we do wholeheartedly agree that access to this kind of continuous measurement equipment will be a limiting factor for future studies and find the idea of creating a basic protocol for gathering a minimum data set extremely intriguing and useful for the field. Hence, as suggested, we will include suggestions for such a minimum data set required for future studies in the revised version of the manuscript.

---

## Author Response (AR2)

**Response to Associate Editor**

It is my pleasure to accept your manuscript in its present form. I just had trouble finding your data set in the Mendeley Data Base using the keywords you provided: license CC BY 4.0 (Knappe et al., 2021). It would be better to provide a direct link (DOI) before publication.

The full Mendeley data doi link has now been added to the paper.